# Trisomy 21 induces pericentrosomal crowding delaying primary ciliogenesis and mouse cerebellar development

Cayla E Jewett[1,2], Bailey L McCurdy[1], Eileen T O'Toole[3], Alexander J Stemm-Wolf[1], Katherine S Given[1], Carrie H Lin[1], Valerie Olsen[1], Whitney Martin[4], Laura Reinholdt[4], Joaquín M Espinosa[2,5], Kelly D Sullivan[2,6], Wendy B Macklin[1], Rytis Prekeris[1], Chad G Pearson[1,2]*

[1]Department of Cell and Developmental Biology, University of Colorado Anschutz Medical Campus, Aurora, United States; [2]Linda Crnic Institute for Down Syndrome, University of Colorado Anschutz Medical Campus, Aurora, United States; [3]Molecular, Cellular, and Developmental Biology, University of Colorado Boulder, Boulder, United States; [4]Jackson Laboratory, Bar Harbor, United States; [5]Department of Pharmacology, University of Colorado Anschutz Medical Campus, Aurora, United States; [6]Department of Pediatrics, Section of Developmental Biology, University of Colorado Anschutz Medical Campus, Aurora, United States

*For correspondence:
Chad.Pearson@cuanschutz.edu

Competing interest: The authors declare that no competing interests exist.

**Abstract** Trisomy 21, the genetic cause of Down syndrome, disrupts primary cilia formation and function, in part through elevated Pericentrin, a centrosome protein encoded on chromosome 21. Yet how trisomy 21 and elevated Pericentrin disrupt cilia-related molecules and pathways, and the in vivo phenotypic relevance remain unclear. Utilizing ciliogenesis time course experiments combined with light microscopy and electron tomography, we reveal that chromosome 21 poly-ploidy elevates Pericentrin and microtubules away from the centrosome that corral MyosinVA and EHD1, delaying ciliary membrane delivery and mother centriole uncapping essential for ciliogenesis. If given enough time, trisomy 21 cells eventually ciliate, but these ciliated cells demonstrate persistent trafficking defects that reduce transition zone protein localization and decrease sonic hedgehog signaling in direct anticorrelation with Pericentrin levels. Consistent with cultured trisomy 21 cells, a mouse model of Down syndrome with elevated Pericentrin has fewer primary cilia in cerebellar granule neuron progenitors and thinner external granular layers at P4. Our work reveals that elevated Pericentrin from trisomy 21 disrupts multiple early steps of ciliogenesis and creates persistent trafficking defects in ciliated cells. This pericentrosomal crowding mechanism results in signaling deficiencies consistent with the neurological phenotypes found in individuals with Down syndrome.

## Editor's evaluation

The authors use human trisomy and tetrasomy 21 cell lines and a mouse model to examine the effects of an additional copy of Pericentrin (PCNT) on cell biology, with a focus on ciliation and ciliary Hedgehog signaling. They demonstrate that modestly increased PCNT levels can attenuate ciliogenesis and may result in trisomy 21-associated phenotypes such as cerebellar growth defects. This work advances our understanding of the trafficking defects caused by increased PCNT and has important implications for our understanding of the cellular basis of trisomy 21, a major hereditary human disorder.

**eLife digest** Human cells typically have 23 pairs of structures known as chromosomes. Each chromosome contains a unique set of genes which provide the instructions needed to make proteins and other essential molecules found in the body. Individuals with Down syndrome have an extra copy of chromosome 21. This genetic alteration is known as trisomy 21 and affects many different organs in the body, leading to various medical conditions including intellectual disability, heart defects, and immune deficiencies.

A recent study showed that cells from individuals with Down syndrome had defects in forming primary cilia – structures on the surface of cells which work as signaling hubs to control how cells grow and develop. These cilia defects were in large part due to excess levels of a protein known as Pericentrin, which is encoded by a gene found on chromosome 21. But it is unclear how Pericentrin disrupts cilia assembly, and how this may contribute to the medical conditions observed in individuals with Down syndrome.

To address these questions, Jewett et al. studied human cells that had been engineered to have trisomy 21. The experiments found that trisomy 21 led to higher levels of Pericentrin and altered the way molecules were organized at the sites where primary cilia form. This caused the components required to build and maintain the primary cilium to become trapped in the wrong locations. The trisomy 21 cells were eventually able to rearrange the molecules and build a primary cilium, but it took them twice as long as cells with 23 pairs of chromosomes and their primary cilium did not properly work.

Further experiments were then conducted on mice that had been engineered to have an extra copy of a portion of genes on human chromosome 21, including the gene for Pericentrin. Jewett et al. found that these mice assembled cilia later and had defects in cilia signaling, similar to the human trisomy 21 cells. This resulted in mild abnormalities in brain development that were consistent with what occurs in individuals with Down syndrome.

These findings suggest that the elevated levels of Pericentrin in trisomy 21 causes changes in cilia formation and function which, in turn, may alter how the mouse brain develops. Further studies will be required to find out whether defects in primary cilia may contribute to other medical conditions observed in individuals with Down syndrome.

## Introduction

Trisomy 21 or Down syndrome (DS) is a common chromosomal disorder characterized by phenotypes including craniofacial abnormalities, intellectual disability, heart defects, and cerebellar hypoplasia (*Haydar and Reeves, 2012*; *Bergström et al., 2016*; *Richtsmeier et al., 2000*). These pathologies overlap with those of ciliopathies—genetic disorders affecting primary cilia (*Goetz and Anderson, 2010*). Primary cilia are signaling organelles essential for vertebrate development (*Goetz and Anderson, 2010*). The primary cilium nucleates from the mother centriole and projects into the extra-cellular space. Cilia formation requires the spatial and temporal coordination of many molecules. It begins with remodeling the mother centriole, including the addition of appendages that serve as docking platforms for cargo delivery (*Schmidt et al., 2012*; *Tanos et al., 2013*; *Ishikawa et al., 2005*). This triggers removal and proteasomal degradation of the mother centriole capping proteins, allowing extension of axoneme microtubules and membrane remodeling to ensheath the axoneme with a ciliary membrane (*Spektor et al., 2007*; *Lu et al., 2015*; *Nachury et al., 2007*). A barrier complex forms at the base of the cilium called the transition zone, that restricts access of molecules to and from the cilium. This creates a unique ciliary compartment that is biochemically distinct from the rest of the cell, allowing the cilium to function as a specialized signaling organelle (*Garcia-Gonzalo and Reiter, 2012*).

Building and maintaining cilia requires trafficking of molecules to and from the centrosome (*Sung and Leroux, 2013*; *Nachury et al., 2010*). The centrosome is comprised of the mother and daughter centrioles and surrounding pericentriolar material. Molecules are trafficked in both membrane-derived vesicles and granular moieties, called centriolar satellites, that move along microtubules nucleated and organized by the centrosome (*Kim et al., 2008*; *Blacque et al., 2018*; *Hori and Toda, 2017*). Peri-centrin (PCNT) is an essential centrosome scaffolding protein that, together with CDK5RAP2/CEP215

and other centrosome components, organizes microtubules (*Doxsey et al., 1994*; *Gavilan et al., 2018*). The *Drosophila* PCNT ortholog is required for cilia function in sensory cells and sperm, and mis-localization of PCNT is sufficient to redistribute the pericentriolar material and microtubules (*Martinez-Campos et al., 2004*; *Galletta et al., 2020*). Furthermore, in human cells, PCNT depletion disrupts primary cilia formation (*Jurczyk et al., 2004*). *PCNT* is encoded on human chromosome 21 (human somatic autosome 21; HSA21), and we previously showed that elevated PCNT due to increased copy number in trisomy 21 is sufficient to initiate cilia defects in human DS-derived fibroblasts compared to age- and sex-matched controls (*Galati et al., 2018*). Elevated PCNT forms aggregates that colocalize with centriolar satellite proteins such as PCM1 and disrupts the flux of intracellular components to and from the centrosome (*McCurdy et al., 2022*). While we showed reduced centrosome localization of the ciliary IFT20 protein in trisomy 21 cells (*Galati et al., 2018*), IFT20 recruitment to the cilium is a late step in the process of ciliogenesis (*Lu et al., 2015*; *Joo et al., 2013*). Thus, it is unknown which molecules or pathways are disrupted in trisomy 21 that reduce cilia formation and signaling, and whether cilia defects observed in cultured cells contribute to DS-associated phenotypes in vivo.

Here, we use isogenic human cell lines to eliminate genetic variability and mouse models of DS to show that elevated PCNT induces trafficking defects around the centrosome such that cargo required for early steps in primary ciliogenesis, including ciliary vesicle formation and mother centriole uncapping, are held up in a 'pericentrosomal crowd'. Of the trisomy 21 cells that do ciliate, intracellular trafficking defects persist as transition zone proteins are unable to reach the centrosome efficiently, thereby decreasing their localization at the transition zone. Consistent with transition zone defects, Shh signaling is reduced and anticorrelates with PCNT levels. A mouse model of DS with increased *Pcnt* copy number and elevated PCNT levels has reduced primary cilia in both primary mouse embryonic fibroblasts (MEFs) as well as cerebellar neuronal precursor cells in vivo. In line with ciliary assembly and signaling defects, these mice have a thinner external granular layer and fewer neuronal protrusions. Our findings reveal how early events in ciliogenesis are disrupted by a PCNT-overexpression-induced crowding phenotype and that these ciliation and signaling defects have consequences for in vivo brain development in DS.

## Results

### Rapid PCNT and microtubule reorganization upon G0 arrest increases with HSA21 dosage

*PCNT* is encoded on HSA21 resulting in 1.5-fold increased expression in trisomy 21 (*Galati et al., 2018*; *McCurdy et al., 2022*). We previously showed that elevated PCNT protein is sufficient for decreased ciliation in trisomy 21 cells (*Galati et al., 2018*; *McCurdy et al., 2022*). Whereas PCNT normally organizes microtubules emanating from centrosomes, elevated PCNT has three major consequences in trisomy 21 cells (*Figure 1A*): (1) PCNT increases microtubule density around the centrosome; (2) PCNT forms large protein aggregates along these microtubules; and (3) PCNT resides at the ends of cytoplasmic microtubules that are disconnected from the centrosome (*Galati et al., 2018*; *McCurdy et al., 2022*). Consistent with more microtubules, the centrosome scaffolding proteins CDK5RAP2 and γ-tubulin are increased at and around the centrosome in trisomy 21 cells, although unlike PCNT, the whole cell expression levels of these proteins are unchanged (*Galati et al., 2018*). Together, elevated PCNT perturbs trafficking to and from centrosomes by altering microtubule networks and the molecular composition of the pericentrosomal region, thereby hindering the exchange of molecules required for ciliogenesis – a phenotype we have termed pericentrosomal crowding. This model was derived from an analysis of either cycling cells or cells 24 hr after serum depletion (0.5% serum in medium) to induce G0 cell cycle arrest and ciliogenesis (*Galati et al., 2018*; *McCurdy et al., 2022*). Because ciliogenesis requires a coordinated series of trafficking and centrosome remodeling events over a period of several hours, we explored how elevated PCNT and trisomy 21 disrupts these processes.

Using hTERT immortalized human Retinal Pigmented Epithelial (RPE1) cells, we performed a time course during ciliogenesis and analyzed how PCNT and microtubule organization change at the centrosome. In control RPE1 cells with two copies of HSA21 (D21), PCNT levels increased rapidly by approximately 50% near the centrosome (within a 5 μm radius) 2 hr after serum depletion and continued to gradually increase throughout the remaining 48-hr time course (*Figure 1B and C*). Similarly, microtubule density around the centrosome increased and fluctuated between 0 and 8 hr post serum

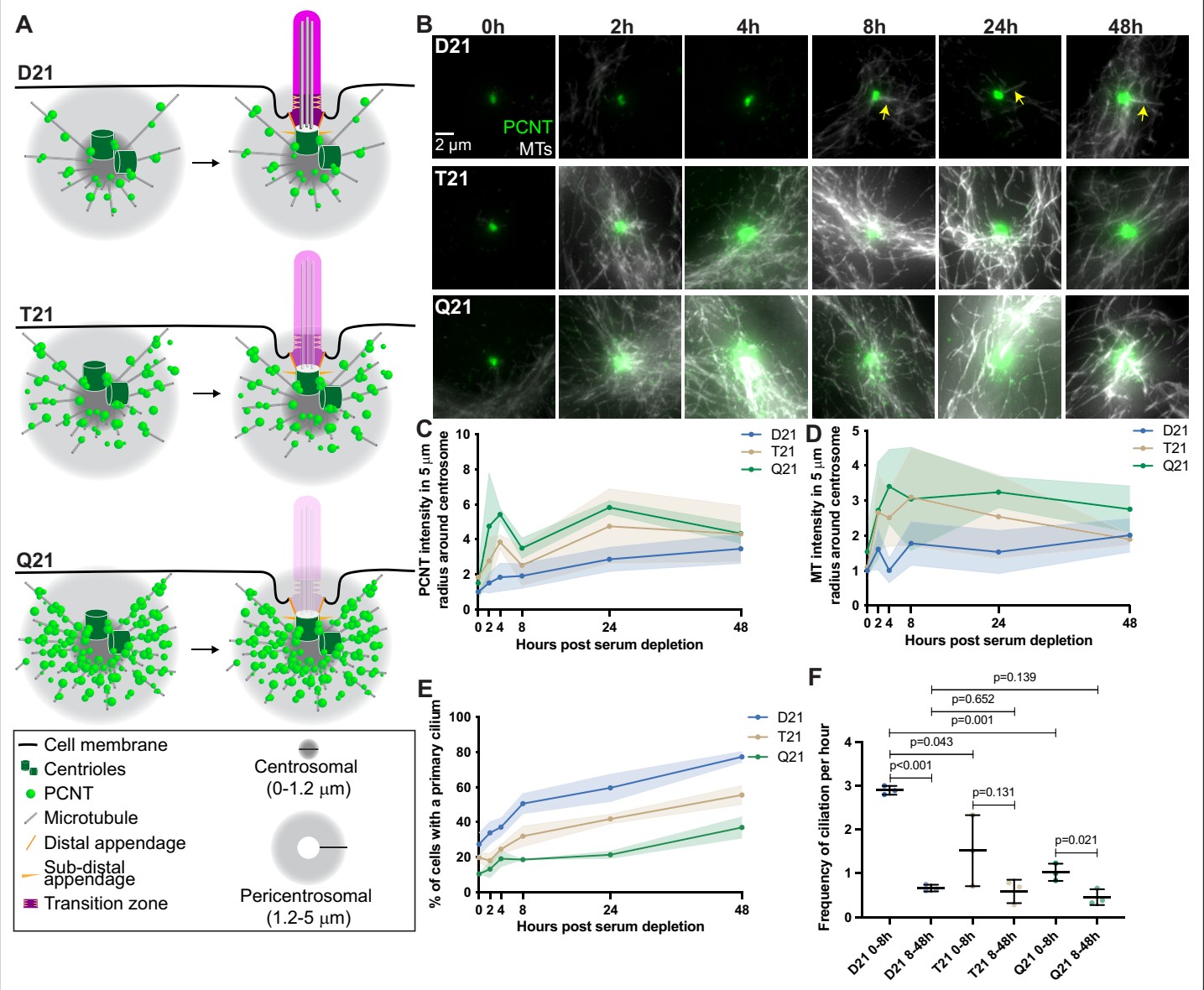

**Figure 1.** Rapid PCNT and microtubule reorganization upon G0 arrest increases with HSA21 dosage. (**A**) Model figure showing the three main consequences of elevated PCNT in trisomy 21 (**T21**) and tetrasomy 21 (**Q21**) compared to disomy 21 (**D21**) cells: (1) PCNT nucleates more microtubules; (2) PCNT forms large protein aggregates on microtubules; and (3) PCNT nucleates microtubules further away from the centrosome. These changes occur predominantly in the pericentrosomal region defined as 1.2–5 μm from the centroid of the centrosome. (**B**) Representative confocal images from time course experiments of RPE1 D21, T21, and Q21 cells grown on coverslips and serum depleted for 0, 2, 4, 8, 24, and 48 hr. Cells were stained with DM1a to label microtubules (MTs) and PCNT. Arrows point to cilium labeled by DM1a staining. (**C**) Quantitation of PCNT intensities in a 5 μm radial circle around the centrosome throughout the time course normalized to D21 average at 0 hr. Graph shows mean ± SD. N=3. (**D**) Quantitation of microtubule intensities in a 5 μm radial circle around the centrosome throughout the time course normalized to D21 average at 0 hr. Graph shows mean ± SD. N=3. (**E**) Quantitation of ciliation frequency throughout the time course using DM1a as a marker for cilia. Graph shows mean ± SD. N=3. (**F**) Quantitation of ciliation frequency per hour calculated by subtracting the percentage of ciliated cells at the starting time point from the percentage of ciliated cells at the ending time point and dividing the result by the total number of hours. Graph shows mean ± SD. N=3. Two-tailed unpaired t-test.

The online version of this article includes the following source data and figure supplement(s) for figure 1:

**Source data 1.** Values for biological and technical replicates for graphs in *Figure 1* and *Figure 1—figure supplement 1*.

**Figure supplement 1.** Rapid PCNT and microtubule reorganization upon G0 arrest increases with HSA21 dosage.

depletion, then remained constant through the rest of the time course (*Figure 1B and D*). To examine the dose dependency of HSA21 and PCNT levels on these processes, we used isogenic human RPE1 cells genetically engineered through microcell-mediated chromosome transfer to have three or four copies of HSA21 (Trisomy 21 /T21 or Tetrasomy 21/Q21) (*Lane et al., 2014*). As expected, Western Blot analysis shows that PCNT protein levels scale with HSA21 dosage, and RNA FISH confirms HSA21 polyploidy (*McCurdy et al., 2022*). T21 and Q21 cells have elevated PCNT prior to serum depletion both at and around the centrosome (within a 5 μm radius) (*Figure 1B and C*). Microtubule intensity was more similar in all three cell lines prior to serum depletion (*Figure 1B and D*). By 2 hr, both PCNT and microtubule intensities elevated with increasing HSA21 dosage (*Figure 1B–D*). Elevated PCNT near the centrosome in T21 and Q21 cells persisted through the time course and was not due to changes in whole cell protein levels (*Figure 1B–D*, *Figure 1—figure supplement 1A–C*). Instead, PCNT and microtubules in T21 and Q21 cells reorganized at the centrosome (0.0–1.2 μm region from the centroid of the centrosome) and pericentrosomal region (1.2–5.0 μm region from the centroid of the centrosome), such that more PCNT foci and microtubules were distributed around the centrosome (*Figure 1B–D*, *Figure 1—figure supplement 1D–G*). These data support a model whereby PCNT accumulates at and around the centrosome upon exit from the cell cycle. Interestingly, by 48 hr, PCNT and microtubule intensities were more similar between the three cell populations (*Figure 1B–D*, *Figure 1—figure supplement 1D–G*), suggesting that T21 and Q21 cells adapt to an elevated PCNT state upon prolonged G0 arrest.

To understand the interplay between PCNT levels, microtubules, and ciliation, we quantified primary cilia frequency through the time course. D21 cells demonstrated two rates of ciliation: an early fast phase from 0 to 8 hr and a late slow phase from 8 to 48 hr (*Figure 1E and F*). In contrast, T21 and Q21 cells showed a decreased ciliation rate in the early phase; however, the slow phase from 8 to 48 hr was similar to D21 cells (*Figure 1E and F*). The early ciliation phase correlates with the increases to PCNT and microtubule intensities which are more robust in T21 and Q21 cells (*Figure 1B–F*). To differentiate between a block or a delay in ciliogenesis, we performed a ciliation time course over 7 days. By day 3 of serum depletion, D21 and T21 cells reach similar ciliation frequencies, and after 4 days all three cell lines reach similar ciliation frequencies (*Figure 1—figure supplement 1H*). This delay in ciliogenesis observed in T21 and Q21 cells is consistent with a model whereby increasing HSA21 dosage disrupts pericentrosomal trafficking flux to and from the centrosome early in the process of ciliogenesis; however, cells can overcome elevated HSA21 dosage to ciliate after a prolonged delay in G0.

## HSA21 ploidy does not affect centriole appendages but decreases vesicles at the mother centriole

Primary ciliogenesis requires the coordination of trafficking and complex assembly events over several hours (*Shakya and Westlake, 2021*). Centriole appendage assembly at the distal end of the mother centriole is a prerequisite for ciliogenesis. Centriole appendages serve as a scaffold for receiving ciliary components trafficked to the centrosome (*Schmidt et al., 2012*; *Tanos et al., 2013*; *Ishikawa et al., 2005*). CEP164 and CEP83 are two distal appendage proteins, with CEP83 adjacent to the centriole microtubule walls and CEP164 at the tip of the appendage structure (*Yang et al., 2018*). In all cell lines (D21, T21, and Q21), CEP164 and CEP83 localized normally at the mother centriole (*Figure 2A*, *Figure 2—figure supplement 1A*). Moreover, the subdistal appendage proteins ODF2 (*Lange and Gull, 1995*; *Nakagawa et al., 2001*) and Ninein (*Mogensen et al., 2000*) localized correctly to the mother centriole (*Figure 2B*, *Figure 2—figure supplement 1B*). Thus, HSA21 ploidy does not affect mother centriole appendages.

To gain further insight into the ultrastructure of centrioles with increasing HSA21 dosage, we performed 3D electron tomography. Consistent with the immunofluorescence data, D21, T21, and Q21 cells showed ninefold symmetry of distal and subdistal appendages (*Figure 2C*). Moreover, the triplet microtubules that comprise the centriole wall were unchanged across all cell lines, confirming that HSA21 ploidy does not affect centriole structure.

Whereas centriole ultrastructure was unaffected, 3D modeling of the EM tomograms revealed changes in the number of vesicles and smooth membranes at the mother centriole. D21 cells showed many small vesicles (35–65 nm diameter) near the mother centriole (91 vesicles), as well as smooth, tubular membranes (*Figure 2D*, red spheres and blue-green objects). In contrast, vesicle number was decreased in T21 and Q21 cells (26 and 31 vesicles, respectively) (*Figure 2D*, red spheres) and smooth

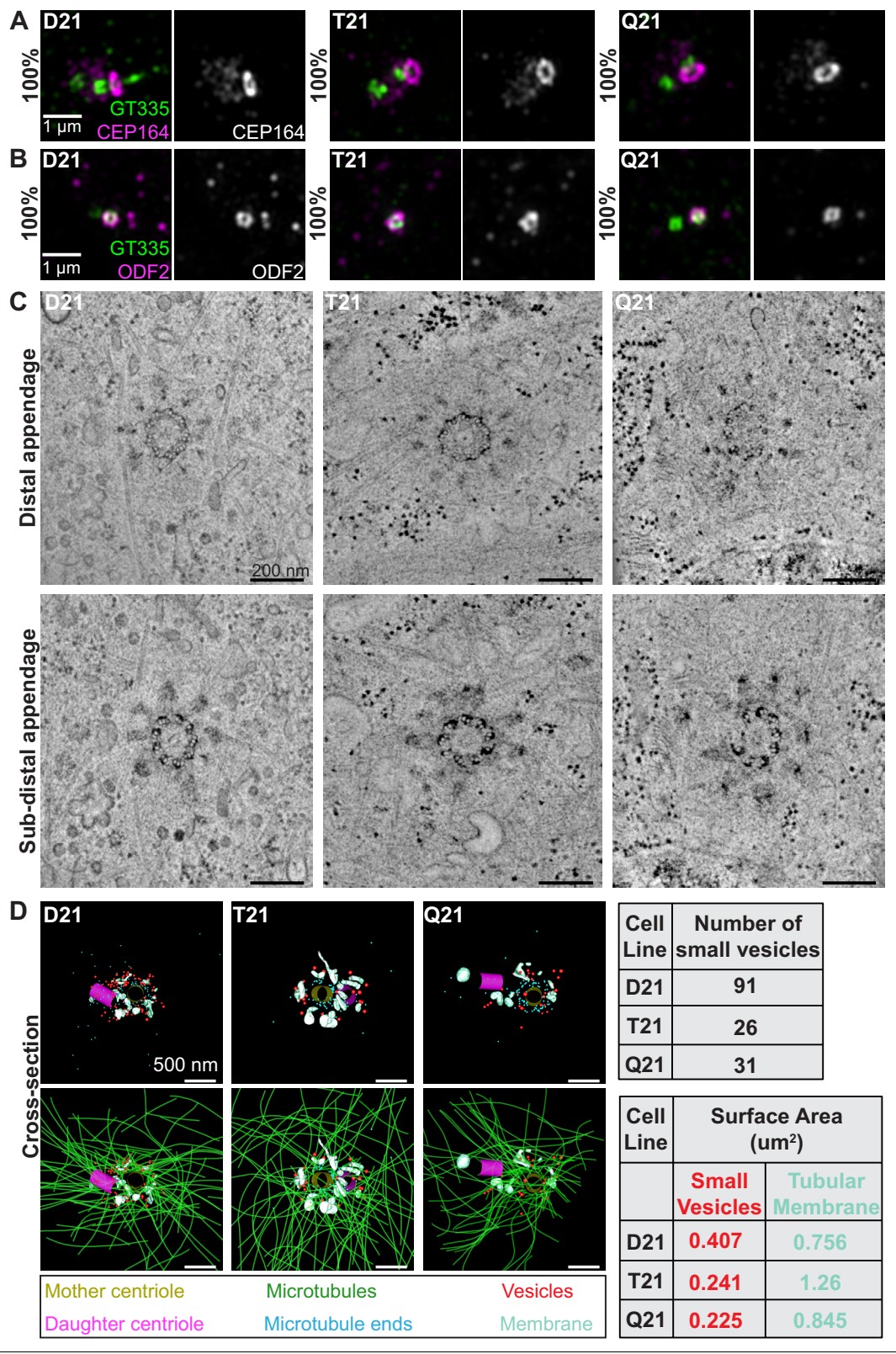

**Figure 2.** HSA21 ploidy does not affect centriole appendages but decreases vesicles at the mother centriole.
(**A–B**) Representative structured illumination microscopy images of RPE1 D21, T21, and Q21 cells grown on
coverslips and serum depleted for 24 hr. Cells were stained with GT335 (centriole and cilia marker) and the distal
appendage marker CEP164 (**A**) or the subdistal appendage marker ODF2 (**B**). Percentages represent cells with

*Figure 2 continued on next page*

*Figure 2 continued*

indicated marker at the mother centriole for 3 N's. (**C**) Selected tomographic slices of RPE1 D21, T21, and Q21 cells serum depleted for 24 hr showing microtubule triplets and distal appendages (top panel) and subdistal appendages (bottom panel). (**D**) 3D models of structures at the centrosome in electron tomograms. Top row shows mother centriole (yellow), daughter centriole (magenta), microtubule minus ends (cyan spheres), vesicles (red), and smooth tubular membranes (blue-green). Bottom row shows microtubules (green), vesicles (red), and smooth membranes (blue-green) surrounding the centriole pair. Tomograms and models shown are of cells prior to ciliary vesicle formation. Tables on right display quantitation of the number of small red vesicles and smooth tubular membrane surface area measurements for D21, T21, and Q21 cells. N=2 reconstructed cells per cell line. Scale bars are 500 nm. Movies of the complete volume and rotating models can be found in *Figure 2—videos 1–3*.

The online version of this article includes the following video and figure supplement(s) for figure 2:

**Figure supplement 1.** HSA21 ploidy does not affect centriole appendages.

**Figure 2—video 1.** Movie of D21 cell from Figure 2 showing the electron tomogram volume, then model projecting from the image, then the model turning without image.
https://elifesciences.org/articles/78202/figures#fig2video1

**Figure 2—video 2.** Movie of T21 cell from Figure 2 showing the electron tomogram volume, then model projecting from the image, then the model turning without image.
https://elifesciences.org/articles/78202/figures#fig2video2

**Figure 2—video 3.** Movie of Q21 cell from Figure 2 showing the electron tomogram volume, then model projecting from the image, then the model turning without image.
https://elifesciences.org/articles/78202/figures#fig2video3

---

membranes were positioned away from the mother centriole (*Figure 2D*, blue-green objects). Moreover, in D21 cells, vesicles were distributed along microtubules and at their ends (*Figure 2D*, green lines, cyan spheres), whereas vesicles in the T21 and Q21 cells were not always found on microtubules (*Figure 2D*, *Figure 2—videos 1–3*). Despite the increased microtubule density observed in *Figure 1*, the comparable number of microtubule minus ends near the centrioles (86, 69, 90 microtubule ends in D21, T21, Q21, respectively) suggests that the additional microtubules are not associated with the mother centriole. Because the tomograms only capture the centrosomal region (0–1.2 μm from the centroid of the centrosome), we did not observe additional microtubule minus ends which our fluorescent studies suggest originate in the pericentrosomal region distal to the centrioles (*Figure 1D*; *McCurdy et al., 2022*). Together, these data indicate that HSA21 dosage does not affect centriole structure nor centriole appendage formation but may alter membrane structures at and around the centrosome.

## Preciliary vesicle components accumulate with PCNT-induced pericentrosomal crowding

Centriolar appendages are docking sites for vesicles and molecules required for ciliogenesis (*Schmidt et al., 2012*). Our electron tomograms suggested that vesicle accumulation at the mother centriole was defective with increased HSA21 ploidy. Because increased HSA21 ploidy resulted in changes to the pericentrosomal region in early ciliogenesis, one potential explanation for decreased mother centriole vesicles is disrupted trafficking in the pericentrosomal region. We thus examined whether molecules required for initiating ciliogenesis are disrupted in this region. The first molecules that initiate the downstream steps of ciliogenesis are components of the preciliary or distal appendage vesicle and include the motor protein Myosin VA (MYO5A) (*Wu et al., 2018*) and the membrane shaping protein EHD1 (*Lu et al., 2015*). At 2 hr post serum depletion, about half of D21 cells have formed a MYO5A vesicle at the mother centriole (*Figure 3A*). In contrast, T21 and Q21 cells show decreased MYO5A vesicle formation and a striking buildup of MYO5A protein in the pericentrosomal region surrounding the centrosome (*Figure 3A and B*). These changes in pericentrosomal MYO5A intensity were not due to changes in whole cell MYO5A protein levels (*Figure 3—figure supplement 1A–C*). Radial analysis of MYO5A intensity surrounding the centrosome in D21 cells showed high MYO5A levels at the centrosome that then decreased in intensity moving away from the centrosome (*Figure 3B* inset). We then compared the MYO5A intensity distribution at and around the centrosome with increasing ploidy through the time course. Two hours after serum depletion, MYO5A intensities in T21 and Q21 cells were increased in the pericentrosomal 1.2–5.0 μm region from the centroid of the

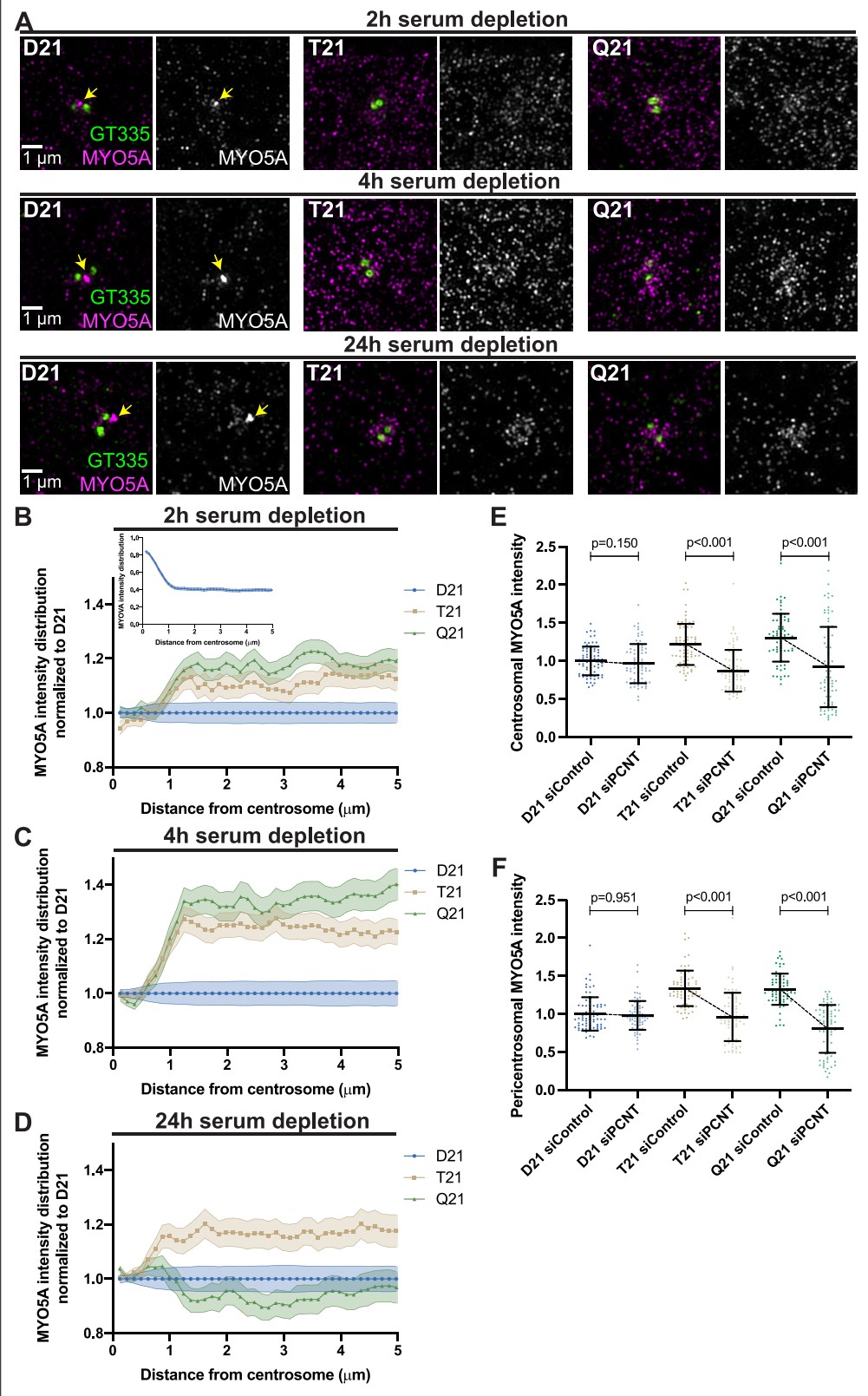

**Figure 3.** Preciliary vesicle components contribute to PCNT-induced pericentrosomal crowding. (**A**) Representative structured illumination microscopy images from time course experiments of RPE1 D21, T21, and Q21 cells grown on coverslips and serum depleted for 2, 4, and 24 hr. Cells were stained with GT335 to label centrioles and MYO5A. (**B–D**) Distribution of MYO5A intensities around the centrosome for 2 (**B**), 4 (**C**), and 24 (**D**) hr timepoints.

*Figure 3 continued on next page*

*Figure 3 continued*

All values were normalized to D21 at 0 µm. Inset in (**B**) shows MYO5A intensity distribution in D21 cells prior to normalization. Graphs show mean ± SD. N=3. (**E**) Quantitation of centrosomal MYO5A intensity in 0–1.2 µm region around the centrosome for control and *siPCNT* treated D21, T21, and Q21 cells. Cells were treated with siControl or *siPCNT* for 24 hr concurrent with serum depletion. All values were normalized to the D21 siControl average. Graph show mean ± SD. N=3. Mann-Whitney U test. (**F**) Quantitation of pericentrosomal MYO5A intensity in 1.2–5 µm region around centrosome for control and siPCNT treated D21, T21, and Q21 cells. Cells were treated with siControl or *siPCNT* for 24 hr concurrent with serum depletion. All values were normalized to the D21 siControl average. Elevated MYO5A levels are distinct from the reduction observed in the distribution analyses (**D**) and are likely the result of the unique conditions for each experiment. Graph show mean ± SD. N=3. Mann-Whitney U test.

The online version of this article includes the following source data and figure supplement(s) for figure 3:

**Source data 1.** Values for biological and technical replicates for graphs *Figure 3* and *Figure 3—figure supplement 1*.

**Figure supplement 1.** Preciliary vesicle components contribute to PCNT-induced pericentrosomal crowding.

**Figure supplement 1—source data 1.** Uncropped blots and protein gels related to *Figure 3—figure supplement 1B*.

---

centrosome (*Figure 3B*). Moreover, centrosomal MYO5A levels were decreased with increasing ploidy (*Figure 3—figure supplement 1D*). By 4 hr, pericentrosomal MYO5A intensities in T21 and Q21 cells became more prominent, while MYO5A centrosomal levels remained decreased (*Figure 3A and C*, *Figure 3—figure supplement 1E*). By 24 hr, T21 and Q21 cells still showed changes in their MYO5A intensity distributions, but the increased intensities were nearer to the centrosome (*Figure 3A and D*, *Figure 3—figure supplement 1F*). Whereas MYO5A intensities increased at the centrosome in T21 and Q21 cells 24 hr post serum depletion, MYO5A appeared more diffuse at the mother centriole compared to D21 cells (*Figure 3A*, bottom panel). This suggests an additional defect in vesicle coalescence. Given that our electron tomography showed decreased vesicles at the centrosome with increasing ploidy, it is likely that the redistribution of MYO5A further away from the centrosome in T21 and Q21 cells contributes to this decrease. Taken together, MYO5A increases in the pericentrosomal region and its accumulation at the mother centriole is hampered with increasing chromosome 21 ploidy, a redistribution consistent with a pericentrosomal crowding model.

We then asked whether elevated PCNT found with increased HSA21 ploidy was sufficient to increase MYO5A in the pericentrosomal region. We have previously shown that reducing PCNT levels in T21 and Q21 cells with siRNA rescues ciliation (*McCurdy et al., 2022*). Reducing PCNT in T21 and Q21 cells to D21 levels with siRNA at 24 hr post serum depletion reduced the centrosomal and pericentrosomal MYO5A to D21 levels (*Figure 3E and F*, *Figure 3—figure supplement 1G, H*). Moreover, the increase in pericentrosomal MYO5A correlates with PCNT levels within individual cells (*Figure 3—figure supplement 1I, J*). Thus, elevated PCNT induces pericentrosomal crowding where MYO5A accumulates during early ciliogenesis, thereby preventing efficient MYO5A vesicle formation at the mother centriole.

We also examined EHD1, an early vesicle protein that is recruited to the mother centriole for coalescence and fusion of preciliary or distal appendage vesicles (*Lu et al., 2015*). We observed decreased EHD1 accumulation at the mother centriole with increasing ploidy and a slight increase in EHD1 in the pericentrosomal region (*Figure 3—figure supplement 1K*). Together, our data support a model whereby elevated PCNT in trisomy 21 accumulates at and around the centrosome immediately after induction of ciliogenesis and induces pericentrosomal crowding, disrupting multiple trafficking pathways required for early preciliary vesicle formation.

## Increased HSA21 ploidy disrupts mother centriole uncapping in a PCNT-dosage-dependent manner

Delivery of preciliary vesicles occurs just prior to or coincident with mother centriole uncapping (*Wu et al., 2018*), which involves removal and degradation of the centriole capping proteins CP110 and CEP97 (*Spektor et al., 2007*; *Liu et al., 2021*). Both CP110 and CEP97 removal from mother centrioles is defective in T21 and Q21 cells compared to D21 cells (*Figure 4A*, *Figure 4—figure supplement 1A–C*). To determine if these uncapping defects were due to elevated PCNT levels, we decreased PCNT levels in T21 and Q21 cells back to D21 levels and examined uncapping. Strikingly,

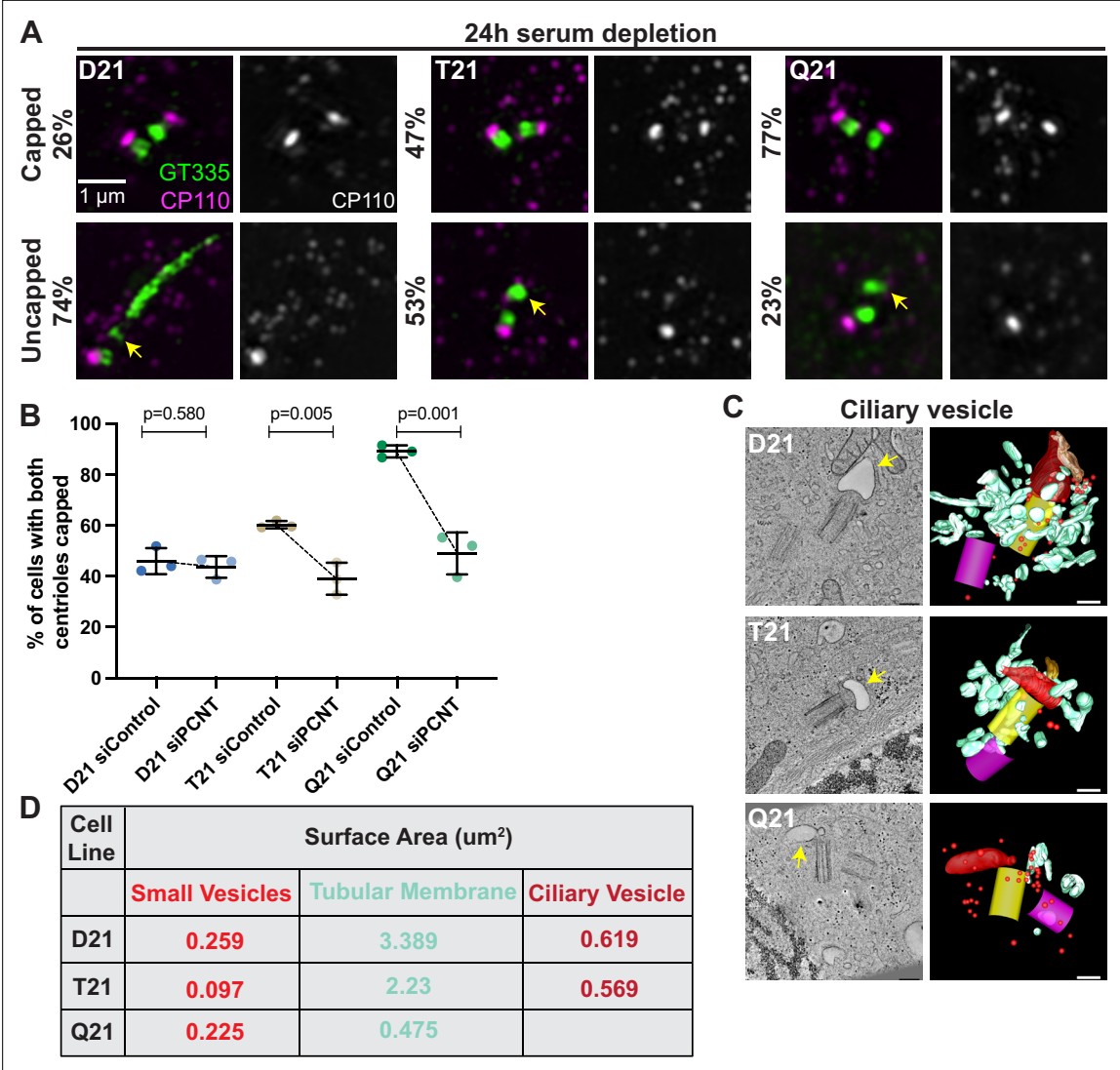

**Figure 4.** Increased HSA21 ploidy disrupts mother centriole uncapping in a PCNT-dosage-dependent manner. (**A**) Representative structured illumination microscopy images of RPE1 D21, T21, and Q21 cells grown on coverslips and serum depleted for 24 hr. Cells were stained with GT335 and the centriole capping protein CP110. Percentages represent cells with indicated phenotype across 3 N's. (**B**) Quantitation of CP110 centriole capping for control and *siPCNT* treated D21, T21, and Q21 cells. Graph show mean ± SD. N=3. Two-tailed unpaired t-test. (**C**) Selected tomographic slices and 3D models showing ciliary vesicle formation in D21, T21, and Q21 cells. T21 and Q21 have smaller ciliary vesicles (arrows) and the ciliary vesicle in Q21 cell is offset from the center of the mother centriole. Models show mother centriole (yellow), daughter centriole (magenta) ciliary vesicle (large red structure at the distal end of the mother centriole), smaller vesicles (red spheres), and smooth tubular membranes (blue-green). (**D**) Quantitation of membrane surface area at the centrosome in electron tomograms for D21, T21, and Q21 cells. D21 cells showed increased small vesicles (red spheres), smooth membranes (blue-green), and ciliary vesicle membrane (red structures) compared to T21 and Q21 cells. Ciliary vesicles in Q21 cells were not quantified due to low frequency (n of 1) and unusual offset of that ciliary vesicle. N=2 reconstructed cells per cell line. Scale bars are 200 nm. Movies of the complete volume and rotating models can be found in *Figure 4—videos 1–3*.

The online version of this article includes the following video, source data, and figure supplement(s) for figure 4:

**Source data 1.** Values for biological and technical replicates for graphs in *Figure 4* and *Figure 4—figure supplement 1*.

**Figure supplement 1.** Increased HSA21 ploidy disrupts mother centriole uncapping, but not general membrane trafficking.

**Figure 4—video 1.** Movie of D21 cell from Figure 4 showing the electron tomogram volume, then model projecting from the image, then the model turning without image.
https://elifesciences.org/articles/78202/figures#fig4video1

**Figure 4—video 2.** Movie of T21 cell from Figure 4 showing the electron tomogram volume, then model projecting from the image, then the model turning without image.
https://elifesciences.org/articles/78202/figures#fig4video2

*Figure 4 continued on next page*

*Figure 4 continued*

**Figure 4—video 3.** Movie of Q21 cell from Figure 4 showing the electron tomogram volume, then model projecting from the image, then the model turning without image.

https://elifesciences.org/articles/78202/figures#fig4video3

reduced PCNT levels rescued CP110 removal from the mother centriole (*Figure 4B*, *Figure 4—figure supplement 1D*). Together, this indicates that elevated PCNT from increased HSA21 ploidy is sufficient to block mother centriole uncapping. Because uncapping requires the trafficking of the preciliary vesicle, we propose that elevated PCNT-induced crowding traps components required for efficient uncapping.

After mother centriole uncapping, preciliary vesicles fuse to form a ciliary vesicle which becomes the nascent ciliary membrane (*Lu et al., 2015*). We examined ciliary vesicle formation using 3D electron tomography. In D21 cells, the ciliary vesicle had a surface area of 0.619 µm$^2$ (*Figure 4C–D* and *Figure 4—video 1*). While some T21 and Q21 cells assembled a ciliary vesicle, the surface areas were smaller in T21 (0.569 µm$^2$) and not measured in Q21 due to the difficulty in finding ciliary vesicles in the tomograms. Additionally, there were decreased vesicles and membranes around the mother centriole (*Figure 4C–D*, *Figure 4—videos 2 and 3*). This is consistent with a redistribution of membranes further away from the centrosome. In addition, the one ciliary vesicle observed in a Q21 cell was not centered over the distal end of the mother centriole. Taken together, this suggests that defects in early steps of ciliogenesis with increasing HSA21 ploidy persist into the ciliary vesicle stages of ciliogenesis.

The small GTPase RAB8 functions in ciliary membrane extension during ciliogenesis (*Lu et al., 2015*) Consistent with our electron tomograms showing decreased membrane at the centrosome in T21 and Q21 cells, we observed decreased RAB8 accumulation at the mother centriole with increasing HSA21 ploidy at 8 hr post serum depletion. This decrease in RAB8-labeled membrane persisted throughout the time course (*Figure 4—figure supplement 1E, F*). Given that RAB8 associates with Golgi and post-Golgi membranes in the context of ciliogenesis (*Nachury et al., 2007*; *Moritz et al., 2001*), one potential explanation for decreased RAB8 at the centrosome is defective trafficking from the Golgi. However, we did not observe changes in RAB8 intensity at the Golgi with increasing ploidy, suggesting RAB8 leaves the Golgi without incident (*Figure 4—figure supplement 1G, H*). Thus, RAB8 trafficking defects are specific to the centrosomal region and are not generally disrupted intracellularly in T21 and Q21 cells.

Given the observed defects in trafficked proteins required for ciliogenesis reaching the centrosome and the importance of microtubules in trafficking pathways, we wondered if general membrane trafficking pathways were disrupted with increasing HSA21 ploidy. *Cis-* and *trans-*Golgi networks, early endosomes, and lysosomes did not exhibit changes in distribution or morphology in T21 and Q21 cells when compared to D21 cells (*Figure 5—figure supplement 1A–G*). Moreover, whereas mild intensity differences were observed in some of these organelles, they did not follow a consistent trend with increasing ploidy (*Figure 5—figure supplement 1B–K*). Thus, gross disruptions of the endolysosomal trafficking pathways are not apparent and trafficking defects associated with trisomy 21 are focused at and around the centrosome. Collectively, this suggests that ciliogenesis defects in trisomy 21 result from early PCNT-induced crowding around the centrosome that captures preciliary vesicle components thereby preventing mother centriole uncapping and RAB8-axoneme extension.

## Decreased transition zone protein localization in ciliated trisomy 21 cells

Despite early defects in ciliogenesis, elevated PCNT from increasing HSA21 ploidy does not abolish cilia. Approximately 40% of T21 and 20% of Q21 cells formed a primary cilium by 24 hr post serum depletion (*Figure 1E*). We thus wondered whether trisomy 21 induced solely a kinetic block in cilia formation or if defects persist after ciliation. To function as a signaling compartment, primary cilia are exposed to the external cellular environment to send and receive signals. In RPE1 cells, cilia typically assemble inside the cell and then fuse with the cell membrane to become extracellular signaling organelles (*Lu et al., 2015*; *Sorokin, 1962*; *Mazo et al., 2016*). This requires remodeling of the plasma membrane and cytoskeletal networks. Given the observed pericentrosomal trafficking defects, we tested whether cilia in trisomy 21 cells were exposed to the external environment using the IN/OUT assay (*Kukic et al., 2016*). No difference was observed in the number of cilia outside versus

inside the cell in D21 and T21 cells (*Figure 5—figure supplement 1L–M*), suggesting that trisomy 21 does not cause cilia to be retained inside cells.

A second requirement for cilia signaling is the creation of a diffusion barrier between the cilium and the cell cytoplasm called the transition zone (*Garcia-Gonzalo and Reiter, 2012*). The transition zone, at the base of the cilium, is composed of proteins that regulate entry and exit of molecules to and from the cilium. Regulation of transit through the transition zone is essential for proper ciliary signaling, and many genes mutated in ciliopathies encode transition zone proteins (*Tobin and Beales, 2009*; *Sang et al., 2011*). Transition zone protein levels were analyzed in ciliated D21 and T21 cells. Q21 cells were eliminated because of their low ciliation frequency. Consistent with defects in the primary cilia transition zone, the core transition zone protein CEP290 (*Craige et al., 2010*; *Yang et al., 2015*) was decreased in T21 cells compared to D21 cells (*Figure 5A*, yellow arrows, and B). In addition to the transition zone, CEP290 also localizes to centriolar satellites (*Kim et al., 2008*), and we observed increased CEP290 at the pericentrosomal but not centrosomal region in T21 cells relative to D21s (*Figure 5A and C*, *Figure 5—figure supplement 1N*). Moreover, CEP290 partially colocalizes with PCNT in this pericentrosomal region (*Figure 5A*, cyan arrows), suggesting that decreased CEP290 at the transition zone may result from pericentrosomal crowding. In addition to CEP290, RPGRIP1L, and two outer transition zone proteins, NPHP4 and TMEM67 *Williams et al., 2011*, were also decreased at the transition zone in T21 compared to D21 cells (*Figure 5D–I*). While we were unable to resolve the structure of the transition zone by EM tomography, we did observe a continued decrease in the amount of membrane at the mother centriole after the primary cilium assembled in T21 cells (*Figure 5J–K*, *Figure 5—videos 1 and 2*). Collectively, trisomy 21 cells show transition zone defects suggesting that trafficking problems to the centrosome are not specific to building a cilium but persist even in ciliated cells. This may explain the signaling defects found previously in cells with elevated PCNT (*Galati et al., 2018*).

## Shh signaling is defective in primary mouse fibroblasts with elevated PCNT

The barrier function of the transition zone is critical for proper ciliary signaling (*Sang et al., 2011*; *Garcia-Gonzalo et al., 2011*) Because transition zone defects were found in trisomy 21 cells, we next asked whether trisomy 21 negatively affects primary cilia dependent signaling. Shh signaling is the best understood cilia-dependent signaling pathway and is important for developmental events that are impacted in DS. Moreover, Shh signaling was previously found to be disrupted in a DS mouse model (*Roper et al., 2006*). Because RPE1 cells lack the tools to study ciliary Shh signaling, we transitioned to primary mouse embryonic fibroblasts (MEFs) derived from DS mouse models. MEFs are commonly used for Shh signaling studies. HSA21 maps to syntenic regions of three mouse chromosomes: MMU10, MMU16, and MMU17 (*Gupta et al., 2016*). We analyzed three mouse models harboring segmental genomic duplications of these syntenic regions on MMU10, MMU16, or MMU17 called Dp10, Dp16, or Dp17, respectively (*Yu et al., 2010b*). The murine *Pcnt* gene is located on MMU10, therefore only the Dp10 model contains an extra copy of the *Pcnt* gene, although other cilia and centrosome-related genes can be found on MMU16 and MMU17 (*Figure 6A*; *Galati et al., 2018*; *van Dam et al., 2013*).

We first asked whether MEFs isolated from Dp10, Dp16, or Dp17 embryos showed ciliation defects. Strikingly, only the Dp10 MEFs with elevated PCNT showed decreased primary cilia frequency when compared to wild-type littermates (*Figure 6B–E*). Cilia frequency in Dp16 and Dp17 MEFs was identical to wild-type (*Figure 6C*, *Figure 6—figure supplement 1A, B*). Dp10 mice contain approximately 41 duplicated HSA21 gene orthologs (*Yu et al., 2010b*) and we previously tested the other cilia and centrosome human orthologs on MMU10 and found no changes in ciliation (*Galati et al., 2018*). Thus, decreased ciliation is specific to elevated PCNT in primary MEFs from the Dp10 mouse model of DS.

Upon induction of the Shh pathway, the transmembrane protein Smoothened (SMO) translocates into the cilium and generates signals that induce nuclear GLI localization and downstream transcriptional responses of hedgehog target genes important for mitogenic activity and developmental processes such as patterning and limb development (*Goetz and Anderson, 2010*; *Kong et al., 2019*). Because Shh signaling is disrupted in a DS mouse model (*Roper et al., 2006*), and we previously found decreased GLI expression in DS-derived human fibroblasts (*Galati et al., 2018*), we examined Shh signaling in MEFs where we can distinguish the contributions of different regions of HSA21. We

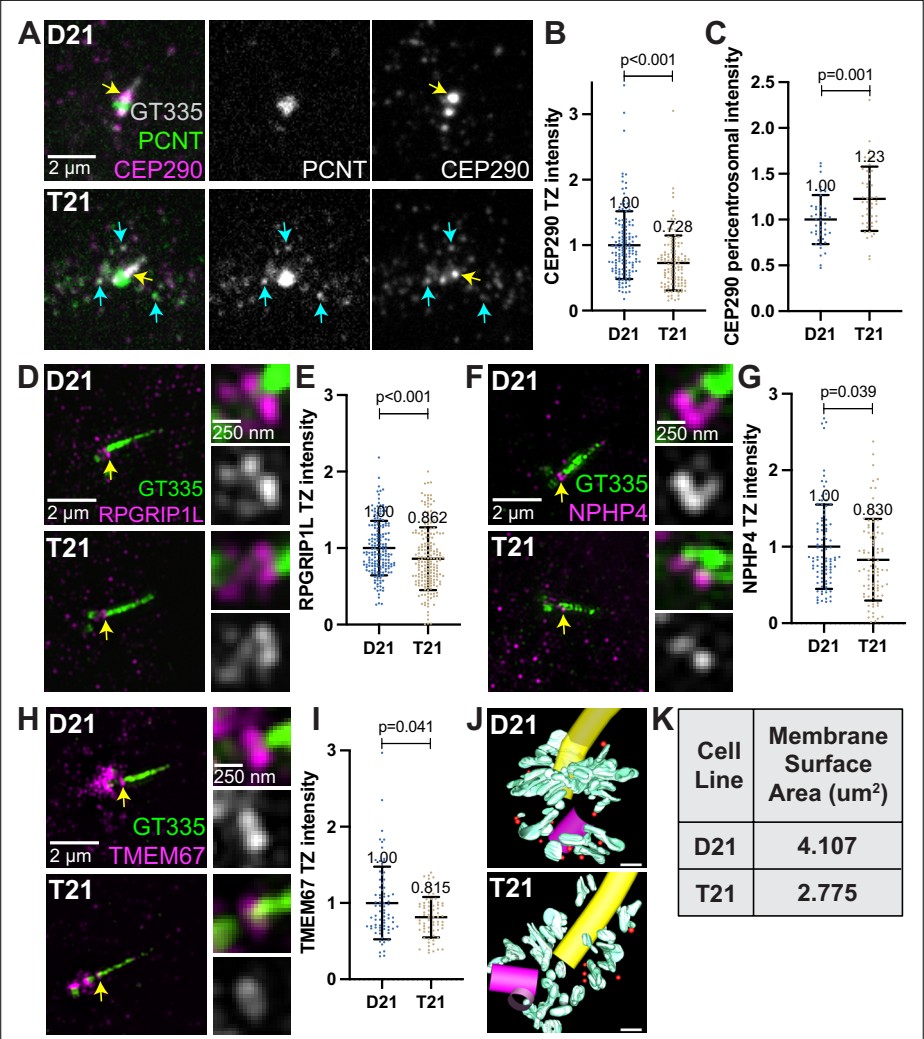

**Figure 5.** *Decreased transition zone protein localization in ciliated trisomy 21* cells. (**A**) Representative confocal images of RPE1 D21 and T21 cells grown on coverslips and serum depleted for 24 hr. Cells were stained with GT335, PCNT, and the transition zone protein CEP290. Yellow arrows point to the CEP290 transition zone population and cyan arrows point to CEP290 satellites that colocalize with PCNT. (**B**) Quantitation of CEP290 transition zone intensity. Graph show mean ± SD. N=3. Mann-Whitney U test. (**C**) Quantitation of pericentrosomal CEP290 intensity in the 1.2–5 µm region around the centrosome. Graph show mean ± SD. N=3. Mann-Whitney U test. (**D, F, H**) Representative structured illumination microscopy images of RPE1 D21 and T21 cells grown on coverslips and serum depleted for 24 hr. Cells were stained with GT335 and the transition zone proteins RPGRIP1L (**D**), NPHP4 (**F**), or TMEM67 (**H**). (**E, G, I**) Quantitation of indicated transition zone protein intensities from confocal images. All values were normalized to the D21 average. Mean intensity values are indicated on graphs. Graphs show mean ± SD. N=3. Mann-Whitney U test. (**J**) 3D models from D21 and T21 cells that contained a primary cilium. An increase in vesicles (red spheres) and tubular membranes (blue-green) at the distal end of the mother centriole is evident in the D21 cell. A procentriole was observed at the T21 daughter centriole (violet). Scale bars are 200 nm. Movies of the complete volume and rotating models can be found in *Figure 5—videos 1 and 2*. (**K**) Quantitation of total membrane surface area at the centrosome in electron tomograms for D21 and T21 cells. N=1 reconstructed cell per cell line.

The online version of this article includes the following video, source data, and figure supplement(s) for figure 5:

**Source data 1.** Values for biological and technical replicates for graphs in *Figure 5* and *Figure 5—figure supplement 1*.

**Figure supplement 1.** General membrane trafficking and ciliary protrusion are not affected by trisomy 21.

**Figure 5—video 1.** Movie of D21 cell with a primary cilium from Figure 5 showing the electron tomogram volume, then model projecting from the image, then the model turning without image.

*Figure 5 continued on next page*

compared SMO intensity in the cilium with and without induction of Shh signaling using the SMO agonist SAG. Without SAG, ciliary SMO levels were undetectable in both wild-type and Dp10 MEFs (*Figure 6F*, *Figure 6—figure supplement 1C*). However, upon SAG treatment, SMO robustly accumulated in the cilium in wild-type cells, whereas ciliary SMO remained low in Dp10 cells (*Figure 6G and H*, *Figure 6—figure supplement 1C*). We next analyzed *Gli1* expression by RT-PCR. Since Gli1 expression prior to SAG treatment was variable, we focused our analysis on cells with low basal *Gli1* expression. In WT cells, SAG treatment increased *Gli1* levels by approximately threefold, whereas *Gli1* levels in Dp10 cells showed little response to SAG treatment (*Figure 6—figure supplement 1F, G*). Moreover, ciliary SMO levels in Dp16 and Dp17 cell lines after SAG treatment inversely correlated with PCNT levels, as Dp16 MEFs had slightly elevated PCNT and slightly decreased ciliary SMO while Dp17 MEFs had slightly decreased PCNT levels and slightly increased ciliary SMO (*Figure 6D, E and*

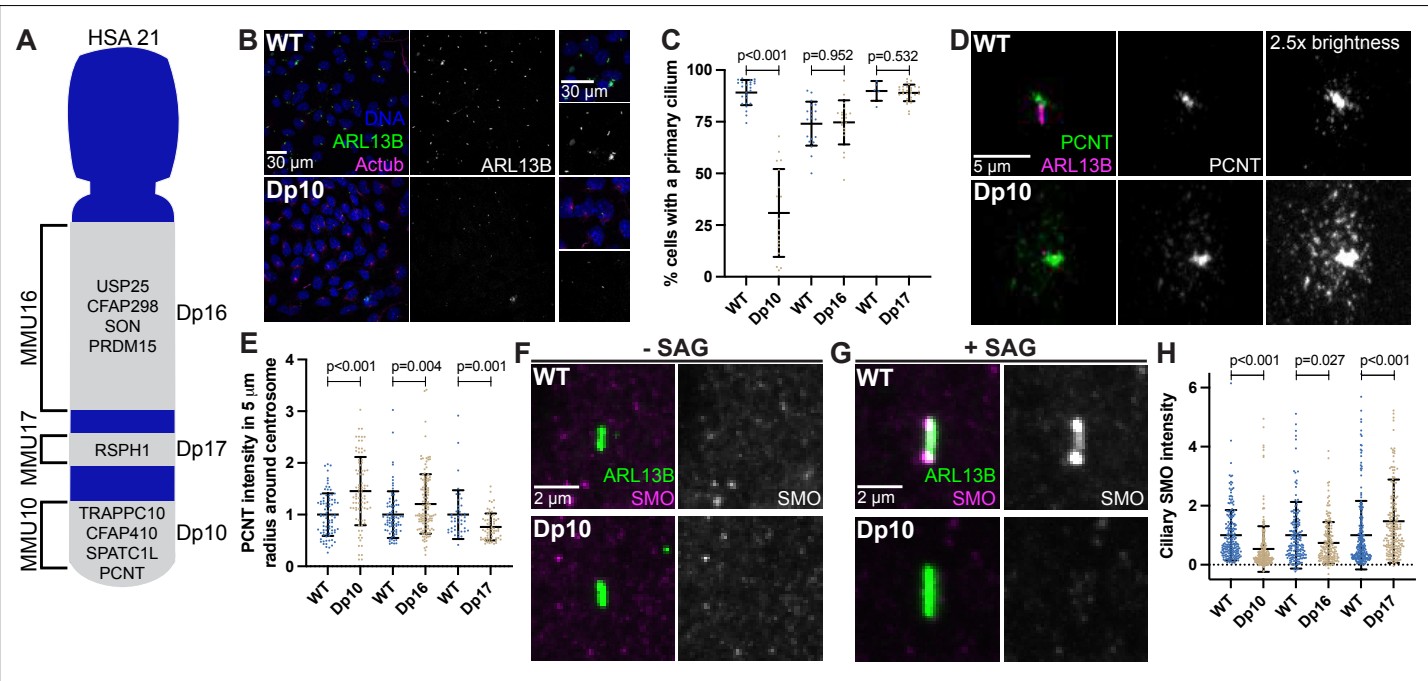

**Figure 6.** Shh signaling is defective in primary mouse fibroblasts with elevated PCNT. (**A**) Cartoon model depicting mouse syntenic regions with HSA21 and corresponding Dp10, Dp16, and Dp17 mouse models. PCNT is located on MMU10. Other cilia and centrosome proteins are also listed. For all following experiments, WT and Dp primary MEFs were isolated from E12.5 pregnant females, grown on coverslips, and serum depleted for 24 hr. (**B**) Representative confocal images of WT and Dp10 MEFs. Cells were stained with Hoechst 33342, the ciliary marker ARL13B, and Actub. (**C**) Quantitation of the number of cells with a primary cilium in WT and Dp MEFs. Graph shows mean ± SD. N=3. Mann-Whitney U test. (**D**) Representative confocal images of WT and Dp10 MEFs. Cells were stained with PCNT and ARL13B. (**E**) Quantitation of PCNT intensities in a 5 μm radial circle around the centrosome in WT and Dp MEFs. Values were normalized to the WT average. Graph shows mean ± SD. N=3. Mann-Whitney U test. (**F, G**) Representative confocal images of WT and Dp10 MEFs untreated (**F**) or treated with 100 nM SAG for the last 4 hr of 24 hr serum depletion (**G**). Cells were stained with SMO and ARL13B. (**H**) Quantitation of ciliary SMO levels in SAG-treated cells for WT and Dp MEFs. Graph shows mean ± SD. N=3. Mann-Whitney U test.

The online version of this article includes the following source data and figure supplement(s) for figure 6:

**Source data 1.** Values for biological and technical replicates for graphs and uncropped gel images in *Figure 6* and *Figure 6—figure supplement 1*.

**Figure supplement 1.** For all following experiments, WT and Dp primary MEFs were isolated from E12.5 pregnant females, grown on coverslips, and serum depleted for 24 hr.

**Figure supplement 1—source data 1.** Uncropped gels from RT-PCR related to *Figure 6—figure supplement 1F*.

*H*, *Figure 6—figure supplement 1D, E*). Together, decreased ciliation frequency and ciliary SMO in Dp10 MEFs is consistent with persistent trafficking defects from pericentrosomal crowding in ciliated trisomy 21 cells.

Interestingly, one Dp10 MEF line (Dp10-2) did not show decreased ciliation compared to wild-type controls (*Figure 6—figure supplement 1H, I*). Importantly, this line also did not exhibit elevated PCNT levels or defects in ciliary SMO localization upon SAG treatment (*Figure 6—figure supplement 1J, K*). It is unclear whether this Dp10-2 line lost the chromosome duplication containing PCNT or whether cells compensated at the molecular level. Regardless, results from this line reinforce the conclusion that cilia and ciliary signaling defects result from elevated PCNT. In summary, PCNT and ciliary SMO levels anticorrelate and are disrupted in trisomy 21.

## Elevated PCNT in a DS mouse model results in decreased primary cilia and cerebellar dysmorphology

Primary cilia are ubiquitous and essential signaling organelles. They are particularly important during brain development and predicted to be disrupted in DS (*Goetz and Anderson, 2010*; *Ho and Stearns, 2021*). Individuals with DS commonly exhibit cerebellar hypoplasia (*Haydar and Reeves, 2012*), and delayed cerebellar development has been observed in the Ts65Dn mouse model of DS, which harbors a duplication of MMU16 genes similar to that of the Dp16 model along with a duplication of genes from MMU17 that are not syntenic to human HSA21 (*Roper et al., 2006*; *Gupta et al., 2016*). During cerebellar development, neuronal precursor cells in the external granular layer respond to Shh that is released from Purkinje cells (*Smeyne et al., 1995*; *Wechsler-Reya and Scott, 1999*). Shh induces mitogenic activity, and the amplified cells ultimately migrate to the internal granule layer where they become mature neurons (*Wechsler-Reya and Scott, 1999*). Primary cilia are required for neuronal precursor cell amplification and defects in Shh signaling reduce proliferation and disrupt cerebellar development (*Spassky et al., 2008*). Whether T21 and elevated PCNT disrupt primary cilia during brain development in vivo remains unknown. We therefore examined ciliation in the external granular layer of postnatal day 4 (P4) mice. Consistent with our findings in Dp10 MEFs, cerebellar neuronal precursors in Dp10 animals had fewer primary cilia compared to wild-type littermates (*Figure 7A and D*). Moreover, Dp16 and Dp17 animals showed no change in ciliation frequency (*Figure 7B–D*). These data are consistent with our analyses in MEFs suggesting that cilia defects occur in animals with elevated PCNT.

Reduced Shh signaling correlates with morphological changes in the cerebellum, such as decreased width of the external granular layer (*Haldipur et al., 2011*; *Nguyen et al., 2018*). We thus measured the width of the external granular layer and found it decreased in Dp10 animals compared to wild-type littermates (*Figure 7E and F*). Decreased width could result from decreased cell proliferation, so we examined Ki67 staining as a marker for cell proliferation. While we observed no changes in the number of Ki67-positive cells in the external granular layer of Dp10 and wild-type animals at P4 (*Figure 7—figure supplement 1A, B*), we cannot rule out changes in cell proliferation at earlier developmental time points. The source of Shh for neuronal precursors comes from Purkinje cells (*Wechsler-Reya and Scott, 1999*). No loss of Purkinje cells was noted in Dp10 pups (*Figure 7—figure supplement 1C*). Finally, given that elevated PCNT alters microtubule networks in cultured RPE1 cells and that PCNT mutations produce severe neuronal defects including disrupted neuronal migration (*Martinez-Campos et al., 2004*; *Delaval and Doxsey, 2010*; *Endoh-Yamagami et al., 2010*), we visualized doublecortin (DCX), a microtubule binding protein that functions in neuronal migration (*Gleeson et al., 1999*; *Francis et al., 1999*). Dp10 animals showed decreased DCX-labeled cellular protrusions compared to wild-type (*Figure 7—figure supplement 1C*). While not conclusive, this suggests that elevated PCNT alters non-centrosomal microtubules such as those required for neuronal outgrowth (*Nishita et al., 2017*). Taken together, elevated PCNT results in decreased ciliation in the external granular layer of the cerebellum and these cerebella demonstrated morphological changes consistent with decreased Shh signaling and developmental defects.

To determine if decreased ciliation in vivo was a direct result of elevated PCNT levels, we cultured cerebellar slices from P4 pups and treated slices with either control or *PCNT* siRNA. Strikingly, reducing PCNT levels in Dp10 slice cultures rescued ciliation back to control levels (*Figure 7G–I*). Taken together, this suggests that elevated PCNT from trisomy 21 is sufficient to induce ciliation defects during cerebellar development in vivo.

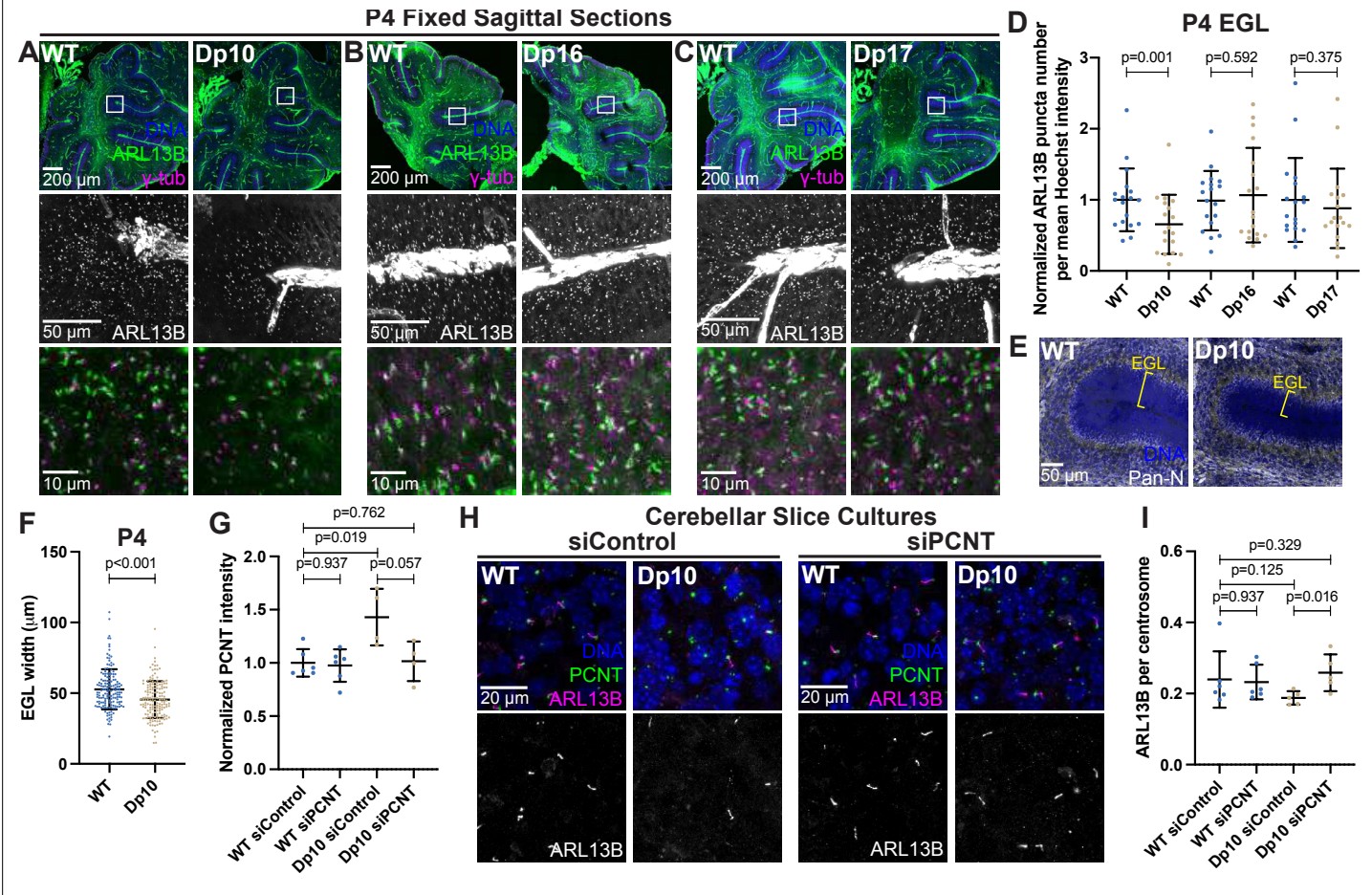

**Figure 7.** Elevated PCNT in a DS mouse model results in decreased primary cilia and cerebellar dysmorphology. (**A–C**) Representative tiled confocal images of the cerebellum from P4 wild-type (WT) and Dp10 (**A**), Dp16 (**B**), and Dp17 (**C**) animals. Brain sections were stained with Hoechst 33342, ARL13B, and γ-tubulin. Insets show progressively zoomed in regions corresponding to the same folia in each animal. (**D**) Quantitation of primary cilia frequency in WT and Dp animals normalized to WT. Graph shows mean ± SD. N=3. Paired t-test. (**E**) Representative tiled confocal images of WT and Dp10 P4 animals corresponding to the same cerebellar folia in each animal. Brain sections were stained with Hoechst 33342 and a Pan-neuronal marker. Yellow bracket denotes external granular layer. (**F**) Quantitation of the external granular layer width in WT and Dp10 animals. Graph shows mean ± SD. N=3. Two-tailed unpaired t-test. (**G**) Quantitation of PCNT intensity in WT and Dp10 cerebellar slice cultures treated with control or PCNT siRNA. Values were normalized to WT siControl averages. Graph shows mean ± SD. N=3. Mann-Whitney U test. (**H**) Representative confocal images of WT and Dp10 cerebellar slice cultures isolated from P4 pups and treated with control or PCNT siRNA for 48 h in serum free media. Slices were stained with Hoechst 33342, ARL13B, PCNT. (**I**) Quantitation of primary cilia frequency in WT and Dp10 cerebellar slice cultures treated with control or PCNT siRNA. Graph shows mean ± SD. N=3. Mann-Whitney U test.

The online version of this article includes the following source data and figure supplement(s) for figure 7:

**Source data 1.** Values for biological and technical replicates for graphs in *Figure 7* and *Figure 7—figure supplement 1*.

**Figure supplement 1.** Normal proliferation and P21 mouse cerebella morphology and defective Doublecortin labeled microtubule extensions.

Our data from T21 and Q21 RPE1 cells suggests that pericentrosomal crowding delays ciliogenesis; however, cells arrested in G0 for a prolonged period eventually ciliate (*Figure 1E*, *Figure 1—figure supplement 1H*). Moreover, our previous work in human DS-derived fibroblasts showed the strongest ciliation defects in cycling cells (*Galati et al., 2018*). We thus tested whether ciliation defects in vivo were dependent on cell cycle state by analyzing cells in the inner granular layer of the cerebellum which are generally post-mitotic. Primary ciliation of inner granule layer cells was unchanged between wild-type and Dp10 animals (*Figure 7—figure supplement 1D*). We also asked whether ciliation defects persisted throughout development in Dp10 animals by analyzing a later developmental time-point. At P21, the cerebella of WT and Dp10 animals showed no difference in ciliation frequency (*Figure 7—figure supplement 1E, F*). Moreover, at P21, the widths of both the cerebellar granule

and molecular layers were unchanged between WT and Dp10 animals (*Figure 7—figure supplement 1G, H*). These findings are consistent with our cultured models of HSA21 ploidy and provide further evidence that trisomy 21 delays but does not abolish ciliogenesis in vivo, with the major impact on cycling cells such as migrating precursor cells.

## Discussion

### Trisomy 21 alters trafficking flux of early ciliogenesis molecules delaying ciliogenesis

Here, we use a time course to establish the dynamic changes to PCNT trafficking puncta and microtubules upon induction of ciliogenesis through media serum depletion. In control euploid cells, the largest changes to PCNT and microtubule networks around the centrosome occur within 2 hr after induction of ciliogenesis. Moreover, we observe a slow and a fast phase to ciliation in a population of cells, with the fast phase occurring within the first 8 hr and the slow phase from 8 to 48 hr. In T21 and Q21 cells, PCNT and microtubule levels increase more dramatically at the onset of ciliogenesis. This is accompanied by a decrease in the effectiveness of the fast phase of ciliogenesis, while the slow phase from 8 to 48 hr is largely intact. Presumably, this rapid increase of PCNT (either by inappropriately nucleating MTs) or sterically hindering trafficking molecules (*McCurdy et al., 2022*) slows trafficking, and in turn slows the rapid phase of cilia assembly. The aberrant microtubule network could result from the PCNT interaction with the microtubule nucleator, γ-tubulin, or through changes to other microtubule nucleators (XMAP215, CDK5RAP2/CEP215, and SAS4/CPAP/CENPJ) (*Roostalu and Surrey, 2017*; *Thawani et al., 2018*). Interestingly, if T21 and Q21 cells experience persistent cell cycle arrest, they will eventually ciliate at similar frequencies to D21 cells. These data reveal that in trisomy 21, PCNT-induced pericentrosomal crowding occurs immediately upon induction of ciliogenesis and delays but does not abolish ciliation.

Several distinct trafficking pathways required for building a primary cilium cannot be efficiently recruited to the site of assembly in the altered microtubule and PCNT landscape caused by increased HSA21 dosage. MYO5A, one of the earliest proteins to localize to the mother centriole during ciliogenesis, does not traffic normally but rather participates in pericentrosomal crowding in T21 and Q21 cells. Mother centriole uncapping is also disrupted, perhaps through defects in trafficking the proteosome or autophagosome machinery required for CP110 and CEP97 removal and degradation (*Spektor et al., 2007*; *Liu et al., 2021*), although this remains to be tested. In addition to early defects in ciliogenesis, pericentrosomal crowding affects later steps such as RAB8-mediated ciliary membrane growth and IFT20 centrosomal localization (*Lu et al., 2015*; *Galati et al., 2018*). Together, our results show HSA21 dosage dependent disruption in trafficking pathways are important to the earliest observable stages of ciliogenesis and cilia function, as molecules required for building and maintaining a cilium get caught in PCNT-induced crowding around the centrosome.

Several lines of evidence support the idea that trisomy 21 delays rather than blocks ciliogenesis. First, with sustained G0 arrest, T21 and Q21 cells increase their ciliation. Prolonged arrest appears to allow for a compensation mechanism that counteracts pericentrosomal crowding as differences in PCNT and microtubule densities at and around centrosomes are reduced over time. Second, a similar phenomenon is observed with cargo, as the amount of pericentrosomal MYO5A decreases over time and MYO5A eventually accumulates at mother centrioles in T21 and Q21 cells. Trisomy 21 cells demonstrate decreased signaling even after a cell builds a primary cilium suggesting that compensation for trafficking defects may be incomplete. This is consistent with previous work showing reduced GLI expression in human trisomy 21 cells and decreased ciliary SMO trafficking in cells overexpressing PCNT (*Galati et al., 2018*). Third, crowding has the most severe consequences for molecules that are dynamically trafficking to and from the centrosome, whereas longer lived structures such as the core centriole and appendages (*Hibbard et al., 2021*) are unaffected by elevated PCNT. Finally, in vivo, only cycling neuronal precursors of the external granular layer have decreased ciliation, whereas ciliation in post-mitotic cells of the inner granular layer and in P21 cerebella remains unaffected with trisomy 21. Collectively, we find that PCNT-induced pericentrosomal crowding disrupts the dynamic flux of molecules to and from the centrosome thereby disturbing the coordination and timing required for proper ciliogenesis and signaling.

## Cerebellar phenotypes in animals with elevated PCNT

While the timing of ciliogenesis is not vital in cultured cells, the timing of ciliogenesis and cilia-dependent signaling is critical during in vivo development where cilia send and receive signals between neighboring cells to coordinate proper tissue development (*Ho and Stearns, 2021*). In line with our cultured cell results, we do not observe a complete loss in cilia and signaling in primary MEFs or cerebellar neuronal precursor cells with elevated PCNT but do observe consistent decreases in ciliation. The organization of cerebellar layers displays a thinner external granular layer in P4 Dp10 mice. While this is suggestive of decreased cell proliferation in cells within the external granular layer, Ki67 staining, a marker for proliferating cells, appears normal in Dp10 animals compared to wild-type littermates. Because elevated PCNT does not result in a complete loss of cilia, perhaps enough of the cell population is ciliated to receive the Shh mitogenic signal and proliferate at this timepoint. Consistent with this, we do not observe gross cerebellar morphology defects or changes in widths of the granule and molecular layers in Dp10 animals at P21 when ciliation is no longer decreased; however, we cannot rule out changes in neuronal processes or glial cells. Cerebellar development is a highly integrated process where a trisomy 21-induced delay in ciliogenesis could alter the coordination of cellular processes such that some cells never fully catch up. Because inputs are not completely lost, this cell-to-cell variability could account for the spectrum of phenotypes observed in individuals with DS.

Most trisomy 21 animal studies have been performed in the Ts65Dn mouse model (*Gupta et al., 2016*). While these mice show some degree of DS-like phenotypes, they are only trisomic for about half of the HSA21 orthologs and contain an additional amplification of genes not found on HSA21 (*Gupta et al., 2016*; *Duchon et al., 2011*). The Dp mouse models are a more refined genetic system to determine the phenotypic contributions from different regions of HSA21. Dp10 mice contain approximately 41 duplicated HSA21 gene orthologs including PCNT while Dp16 and Dp17 mice have 115 and 19 duplicated gene orthologs, respectively, and exclude PCNT (*Yu et al., 2010b*). Our experiments in primary MEFs and cerebellar slices largely attribute cilia defects to elevated PCNT levels in Dp10 animals. However, there are other known cilia and centrosome genes found on HSA21, and potentially non-coding regions that may also contribute to phenotypic consequences of trisomy 21. For example, MMU16 contains the splicing factor SON, which is known to splice PCNT mRNA and alters PCNT levels and distributions (*Stemm-Wolf et al., 2021*; *Ahn et al., 2011*). Indeed, while we do not find changes in primary cilia frequency in Dp16 animals, we do observe moderately decreased ciliary SMO and moderately increased PCNT levels. Animals with individual chromosomal duplications, Dp10, Dp16, or Dp17 *Yu et al., 2010b*, do not show as severe phenotypic consequences as combined Dp10; Dp16; Dp17 animals (*Yu et al., 2010a*). Therefore, while elevated PCNT disrupts ciliogenesis and signaling, there are likely other contributions from additional genes on HSA21 that add to the spectrum of phenotypes observed in individuals with DS.

In contrast to single gene disruption studies, here we show that ciliary defects arise from elevated protein expression, as is often the case in chromosomal aberrations. Indeed, cilia defects have been observed with increased copy number of a nuclear pore protein (*Del Viso et al., 2016*). While changes in some protein levels may be tolerated, we demonstrate that even a modest increase in PCNT protein (1.5–2-fold) is deleterious to cilia formation and function in a dose-dependent manner. Elevated PCNT aggregates and reorganizes microtubule networks, inducing pericentrosomal crowding that disrupts cargo transport required for early steps in ciliogenesis and persists in ciliated trisomy cells. Because tight control of microtubule organization is essential for many cell types and processes including neuron outgrowth, cell migration, immune synapse formation, glucose-induced insulin secretion, and cell polarity, these functions need further study to determine whether elevated PCNT from trisomy 21 and changes in microtubule topologies alter these cell types and functions.

## Materials and methods

### Cell lines

Disomy 21, Trisomy 21, and Tetrasomy 21 (D21, T21, Q21) hTERT-immortalized retinal pigment epithelial (RPE1) cells were generated by Drs. Andrew Lane and David Pellman at the Dana-Farber Cancer Institute (*Lane et al., 2014*). Cells were grown in DMEM:F12 (SH30023; Cytiva) supplemented with 10% fetal bovine serum (FBS, Peak Serum; PS-FB2) and 1% Penicillin/Streptomycin at 37 °C and 5%

$CO_2$. Cells were passaged 1:5 at ~80–90% confluency with 0.25% Trypsin (15090–046; Gibco). Cells were routinely screened for mycoplasma and validated by FISH, RNAseq fits, and PCR. Primary MEFs were generated as described in *Mariani et al., 2016*. In brief, embryonic day 12.5 embryos were dissected from pregnant females. The head and internal organs were removed, and the remaining tissue was dissociated by passage through a 22-gauge needle. The head was used for genotyping. Dissociated cells were cultured in DMEM with 10% fetal bovine serum and 1% Penicillin/Streptomycin for no more than 3 passages.

## Immunofluorescence

Cells were plated on collagen-coated coverslips and grown in media with 10% FBS until 80–90% confluent. To induce ciliogenesis, cells were washed once with 1 x PBS and then grown in serum depleted media (DMEM:F12 with 0.5% FBS) for indicated time. For time course experiments, cells were fixed according to *Waterman-Storer and Salmon, 1997* following 0, 2, 4, 8, 24, or 48 hr of serum depletion. Briefly, cells were pre-permeabilized for 5 min in 0.5% TritonX-100 in PHEM (60 mM PIPES, 25 mM HEPES, 10 mM EGTA, 2 mM $MgCl_2$, 6.9 pH). Cells were then fixed with 4% paraformaldehyde/0.5% glutaraldehyde for 20 min and quenched with 0.1% sodium borohydride. Following quenching, cells were washed four times in 0.1% Triton X-100 in PHEM and stored at 4 °C until immunostaining.

For all other staining experiments, cells were fixed with 4% paraformaldehyde for 20 min at room temperature or with 100% ice cold methanol for 10 min at –20 °C. Cells were blocked for 1–2 hr in blocking buffer (PBS, 0.1% Triton X-100, 10% normal donkey serum). Primary antibodies were diluted in blocking buffer and incubated overnight at room temperature. Cells were washed with PBS with 0.1% Triton X-100 before adding secondary antibodies for 1–2 hr at room temperature. Cells were washed again before mounting in VectaShield (Vector Labs) and sealing with nail polish or mounting in Prolong Gold (ThermoFisher) for structured illumination microscopy experiments. Coverslips for all experiments were #1.5. Primary and secondary antibodies are listed in the key resources table.

For SAG treatment, MEFs were grown on coverslips to 80–90% confluency, moved to serum-depleted media for 20 hr, treated with 100 nM SAG in serum-depleted media for 4 hr, fixed with 4% PFA and stained as described above. The SMO antibody was a gift from Dr. Rajat Rohatgi (Stanford) (*Rohatgi et al., 2007*).

To costain PCNT with other rabbit primary antibodies, PCNT was conjugated to Alexa Fluor 488 (Antibody labeling kit: Invitrogen A20181).

## Fluorescence microscopy

Confocal images were acquired using either a Nikon Eclipse Ti 2 inverted A1 Confocal microscope with Nikon Elements software or Nikon Eclipse Ti inverted microscope equipped with an Andor iXon X3 camera, CSU-X1 (Yokogawa) spinning disk, and Slidebook 6 software. Tiled brain images at P4 were acquired using a Nikon Eclipse Ti 2 inverted A1 Confocal microscope and stitched in the Nikon Elements software. Tiled brain images at P21 were acquired on a SP8 (Leica Microsystems) confocal microscope and stitched in the Leica LAS X software. Super resolution imaging was performed on a Nikon 3D structured illumination microscopy system (Ti 2 Eclipse) with a 100 x TIRF objective (NA 1.45). Images were captured with a complementary metal-oxide semiconductor camera (Orca-Flash 4.0; Hamamatsu). Raw images were reconstructed using the Nikon Elements image stack reconstruction algorithm. Images were processed in FIJI (*Schindelin et al., 2012*). Figures were made in Adobe Illustrator. A minimum of three biological replicates were performed for each experiment unless otherwise noted. All images presented in figures are max projections.

## IN/OUT cilia assay

RPE1 D21, T21, and Q21 cells stably expressing pHluorin-Smoothened (pLVX-pHluorin-Smoothened vector was a gift from Dr. Derek Toomre [Yale University]) were plated on collagen-coated coverslips, fixed and stained as described previously (*Kukic et al., 2016*). Briefly, culture media was removed, cells were gently washed in PBS, and then fixed in 4% PFA in PBS for 10 min. Cells were then blocked in 5% normal donkey serum in PBS for 30 min followed by incubation with anti-GFP primary antibody solution for 1 hr. Cells were fixed again for 10 min, permeabilized with 0.1% Triton-X, and incubated in another primary antibody solution containing anti-Actub for 1 hr. After gentle washing, cells were

incubated with secondary antibodies and Hoechst 33342, followed by washing and mounting on slides.

## RNAi

Human PCNT siRNA (Smart Pool) (M-012172-01-0005; Dharmacon) was transfected into RPE1 cells with Lipofectamine RNAi MAX (13778100; ThermoFisher Scientific) according to the manufacturer's protocol. Mission siRNA universal negative control #1 was used for all negative controls (SIC001-1NMOL; Sigma). All siRNAs were used at a final concentration of 25 nM. Cells were treated with siRNA in serum depleted media for 24 hr before fixation and subsequent immunostaining steps.

## Generation of lentiviral stable cell lines

RPE1 D21, T21, and Q21 lentiviral stable cell lines were generated by transfecting HEK293T cells with lentiviral GFP-EHD1 or pH-Smoothened constructs and lentivirus packaging plasmids using Lipofectamine 2000 (11668-027; Invitrogen). HEK293T media containing virus was collected and added to target cells in the presence of 10 µg/mL polybrene. Transduced cells were selected using 10 µg/mL puromycin for 3 days. To induce GFP-EHD1 expression, 0.125 µg/mL doxycycline was added to serum-depleted media for 24 hr before fixation.

## Western blot analysis

D21, T21, and Q21 RPE1 cells were incubated in starvation media for 24 hr before lysing in modified RIPA buffer containing Benzonase (SLCG0562, Sigma-Aldrich), 5 mM PMSF, 5 mM DTT, and 0.2 mg/mL RNase A. Lysates were separated in 3–8% Criterion XT Tris-Acetate PAGE (3450131, Bio-Rad), and protein transferred onto PVDF membranes (Millipore). Membranes were probed with anti-MYOVA followed by incubation in HRP-conjugated secondary antibody. Blots were incubated for 5 min in SuperSignal West Pico PLUS Chemiluminescent Substrate (34578, ThermoScientific) and imaged on a Bio-Rad ChemiDoc MP Imaging System. For relative MYOVA protein level quantification, band intensities were measured and normalized to india ink loading controls. All band intensity analyses were quantified using ImageJ.

## Gli1 RT-PCR

MEFs were serum depleted for 24 hr in DMEM medium (Cytiva SH30022.01) supplemented with 0.5% fetal bovine serum and treated with 100 nM SAG for the final 4 hr. RNA was isolated from MEFs using the TRIzol Reagent according to the manufacturer's instructions (Thermo Fisher Scientific). 25 units of SuperScript IV (Thermo Fisher Scientific) with 0.1 µM gene-specific primer was used for cDNA first strand synthesis using 500 ng total RNA in a total volume of 10 µl according to the manufacturer's instructions. PCR was performed with *OneTaq* DNA polymerase (New England Biolabs), using 32 amplification cycles. Primers are *Gli*1 F: GAATTCGTGTGCCATTGGGG; *Gli1* R: TGGGATCTGTGTAGCGCTTG; *Pcna* F: GCACGTATATGCCGAGACCT; *Pcna* R: GTAGGAGACAGTGGAGTGGC. Bands were quantified using ImageJ by subtracting background signal from each band and then normalizing Gli1 band intensities to the corresponding Pcna band intensities for each sample. Fold change was calculated by dividing the normalized Gli1 SAG treated intensity by the Gli1 intensity without SAG.

## Mouse models of Down syndrome

Mouse models were obtained from the NICHD funded Cytogenetic and Down Syndrome Models Resource Jackson Laboratory. These include B6.129S7-Dp(16Lipi-Zbtb21)1Yey/J (Dp16; stock# 013530), B6;129-Dp(10Prmt2-Pdxk)2Yey/J (Dp10; stock# 013529), and B6;129-Dp(17Abcg1-Rrp1b)3Yey/J (Dp17; stock# 013531). Strains were maintained on a standard chow diet and a 14 hr light/10 hr dark cycle. Littermate controls were used.

## Cerebellar slice cultures

Cerebella were dissected from postnatal day 4 pups in ice cold Hank's media and sliced 300 µm thick on a tissue chopper (*Liu et al., 2017*). Slices were transferred to MilliCell filter inserts (Millipore PICM03050) only taking intact slices with at least 5 folia. Excess Hank's was removed from filters and replaced with serum-free Neurobasal media supplemented with 1% $N_2$, 0.5% Penicillin/Streptomycin, 0.25% L-glutamine, and 10 mM HEPES. Cerebellar slices were cultured at 37 °C and 5% $CO_2$ and

media was changed 8 hr after plating. Slices were grown for 48 hr before addition of 500 µm control or PCNT siRNA (D-001910-10-05 Accell Non-targeting Pool; Accell Mouse Pcnt 18541 siRNA—SMART-pool) and then cultured for an additional 48 hr before fixation with 4% PFA and staining as described above.

## Mouse brain immunohistochemistry

Experiments were blinded to gender. P4 pups were anesthetized with Isoflurane before decapitation. Tail clips were used for genotyping. Brains were dissected out and fixed in 4% PFA in PBS overnight at 4 °C. Brains were then moved to 30% sucrose in PBS for 1–2 days at 4 °C. Fixed brains were sliced through the midline; each half was embedded in optimal cutting temperature compound (OCT), frozen on dry ice for ~15 minutes, then stored at –80 °C until ready to section. To section, blocks were mounted in OCT and 20 µm sagittal sections were cut using a Leica CM 1950 cryostat micro-tome. Sections were placed on FisherBrand charged slides (Cat # 12-550-15) and stained on slides as described above.

## Genotyping

Tissue from Dp10 pups or embryos were lysed by placing tail clips or heads in Gitschiers Buffer (67 mM Tris pH 8.8, 0.166 mM $(NH_4)_2SO_4$, 6.7 mM $MgCl_2$, 0.005% Triton X-100, 0.1 mg/ml Proteinase K in water) for 2 hr at 55 °C then 95 °C for 10 min. Tissue from Dp16 or Dp17 pups or embryos were lysed by placing tail clips or heads in 50 mM NaOH and heating at 98 °C for 2 hr before neutralizing with 1 M Tris pH 8.0 (1:10). Lysed tissue was used for PCR and the resulting banding pattern from PCR was used to determine genotype. Genotyping primers are as follows: Dp10For: GGCGAACG TGGCGAGAAA; Dp10Rev: CCTGCTGCCAAGCCATCAG; Dp16For: CTGCCAGCC ACTCTAGCTCT; Dp16Rev: AATTTCTGTGGGGCAAAATG; Dp17For: GGAGCCAGGGCTGATGGT; Dp17Rev: CAAC GCGGCCTTTTTACG. Primers for Cux2 were used as controls. Cux2For: GGGACATCACCCACCG GTAATCTC; Cux2Rev: GACCACTGAGTCTGGCAACACG.

## Image analysis

All intensity analysis was performed on max projected images unless otherwise noted. Radial fluorescence intensity was measured using the Radial Profile Extended ImageJ plugin (http://questpharma.u-strasbg.fr/html/radial-profile-ext.html). Briefly, this plugin plots average fluorescence intensity as a function of distance from a user defined centroid. 10 µm diameter ROIs were centered over the brightest pixel of a Gaussian blurred maximum intensity projection of the centrosome defined by PCNT fluorescence. In cells with GT335-labeled centrioles, the centroid was manually centered between the two centrioles. Background was determined per field of view by the mean intensity of an extracellular ROI. Background subtracted radial intensities were summed and normalized to the D21 average. Centrosomal fluorescence intensity was defined as the sum of radial intensities falling within 0.0–1.2 µm from the centroid of analysis. Pericentrosomal fluorescence intensity was defined as the sum of radial intensities falling within 1.2–5.0 µm from the centroid.

Whole cell intensities were calculated using the integrated density within cells outlined using ImageJ. Ciliation frequency was measured by counting either DM1a (*Figure 1*) or ARL13B (*Figure 6*) and dividing by the number of nuclei per field of view. For ciliation frequency in cultured cerebellar slices, the number of ARL13B-labeled cilia was divided by the number of centrosomes per field of view. For ciliation frequency in fixed cerebellar slices, the number of ARL13B-labeled cilia was divided by the mean Hoechst 33342 intensity per field of view because individual nuclei could not be resolved in the thick slices. The frequency of ciliation per hour (*Figure 1*) was calculated by subtracting the percentage of ciliated cells from the previous time point from the percentage of ciliated cells and dividing the result by the total number of hours.

For PCNT siRNA experiments, radial MYO5A and PCNT fluorescence intensity was calculated using the same methods as above. For MYO5A radial distribution, all values were divided by the maximum intensity value per cell and normalized to D21 values. Correlation analyses were performed by plotting 10 µm diameter PCNT intensities normalized within a cell line to either centrosomal or pericentrosomal normalized MYO5A intensities. R values and significance were calculated by running a correlation analysis in GraphPad. PCNT levels were analyzed with a 10 µm diameter circle centered

on the centrosome to measure the integrated density. Background was determined with an extracellular ROI. Background subtracted values were normalized to the D21 average.

GM130 and Golgin97 intensities were measured by drawing an ROI around the GM130 or Golgin97 signal and measuring the integrated density within this ROI using ImageJ. Background was determined by an extracellular ROI. Background subtracted values were normalized to the D21 average. To quantify early endosome and lysosome levels, integrated density was measured across a field of view and divided by the number of nuclei within the same field of view. Values were normalized to the D21 average.

All transition zone antibodies were co-stained with GT335. The transition zone was identified by the gap in GT335 signal between the mother centriole and the base of the cilium. Transition zone protein intensity was determined by centering a 1 μm x 1 μm box over the transition zone and measuring the integrated density. Background was determined by a 1 μm x 1 μm box randomly placed near the centrosome. Background subtracted values were normalized to the D21 average. For TMEM67, the ROI was a 0.5 μm x 0.5 μm box to avoid TMEM67 signal that localizes to the centrosome.

## Statistical analyses

All data sets were tested for Normality using D'Agostino and Pearson Omnibus Normality Test in GraphPad Prism 9. Normal data sets were then tested for significance with a two-tailed unpaired t-test and non-normal data sets were tested for significance using the Mann-Whitney test. For ciliation analysis in mouse cerebella paired t-tests were used to ensure that corresponding folia in WT and Dp10 animals were compared. All graphs show mean ± SD. All experiments utilized at least three biological replicates, unless otherwise noted.

## Electron tomography

RPE1 cells were grown on sapphire discs and prepared for electron microscopy using high pressure freezing and freeze substitution as described in *McDonald et al., 2010*. Briefly, 3 mm sapphire discs (Technotrade International) were coated with gold and a large "F" was scratched into the surface to orient the cell side. The disks were coated with collagen, sterilized under UV light and cells plated for culturing. Monolayers grown on sapphire discs were frozen using a Wohlwend Compact 2 high pressure freezer (Technotrade International). The frozen cells were freeze substituted in 1% OsO4 and 0.1% uranyl acetate in acetone at –80 °C for 3 days then gradually warmed to room temperature. The discs were then flat embedded in a thin layer of Epon resin and polymerized at 60 °C. Regions containing cells were identified in the light microscope, and a small square of resin containing the cells was excised and remounted onto a blank resin block. The cells were then sectioned en face and serial, thick sections (250–300 nm) were collected onto formvar-coated slot grids. Grids were post stained with 2% uranyl acetate and Reynold's lead citrate and 15 nm colloidal gold (BBI International) was affixed to the section surface to serve as alignment markers.

Grids containing serial thick (250–300 nm) sections were screened using a Tecnai T12 microscope (Thermo Fisher Scientific, Waltham, MA). A total of 20 D21 cells were screened, and 4 complete centrosomes were selected for tomographic reconstruction. A total of 26 T21 cells were screened, and 3 complete centrosomes were selected for tomographic reconstruction. A total of 17 Q21 cells were screened and 4 were chosen for reconstruction. Only those cells that had the centrosome complete in the serial sections were selected for reconstruction. However, observations about the presence of ciliary vesicles and primary cilia could be made from the incomplete centrosomes.

Tomography was performed using a Tecnai F30 microscope operating at 300 kV (Thermo Fisher Scientific, Waltham, MA). Dual axis tilt series were collected over a+/-60° range using the SerialEM image acquisition software (*Mastronarde, 2005*), and a Gatan OneView camera (Gatan, Inc, Pleasanton, CA). For most data sets, tilt series were collected from 2 to 4 serial sections to reconstruct a larger volume of cell data. Tomograms were computed, serial tomograms joined and cellular features were modeled using the IMOD 4.9 software package (https://bio3d.colorado.edu/imod/) (*Kremer et al., 1996*; *Mastronarde, 1997*).

Organelles at the centrosomes of the cells (centrioles, vesicles, positions of MT and their ends) were manually traced in these reconstructions using the 3dmod program in the IMOD software package (*Kremer et al., 1996*). Models were projected in 3D to show the arrangement of the centrioles, vesicles, microtubules and the position their ends within the volume. In total, 4 centrosomes from D21, 3

centrosomes from T21, and 4 centrosomes from Q21 were reconstructed and modeled. The modeled centrosomes came from different stages, including at or before ciliary vesicle formation, and one D21 and T21 set with a primary cilium. Surface areas of membrane structures were computed from the model data using the program imodinfo from the IMOD software package.

## Acknowledgements

We thank Drs. Andrew Lane and David Pellman for RPE1 D21, T21, and Q21 cell lines, Dr. Chris Westlake for EHD1 plasmid, Dr. Rajat Rohatgi for SMO antibody, Dr. Tamara Caspary for advice on isolating primary MEFs, and Dr. Derek Toomre for pH-Smoothened plasmid. We are grateful to Drs. Carolyn Ott, Santos Franco, and the Pearson lab for helpful discussions. Electron microscopy was done at the University of Colorado, Boulder EM Services Core Facility in the MCDB Department with Garry Morgan providing specimen preparation. This research was funded by NIH R01GM138415 and R35GM140813 to CGP, NSF Graduate Research Fellowship DGE-1553798, NIH INCLUDE T32 supplement GM008730, and Blumenthal Fellowship to CEJ, and NIH R01DK064380 to RP. CEJ, JME, KDS, and CGP are members of the Linda Crnic Institute for Down syndrome.

## Additional information

### Funding

| Funder | Grant reference number | Author |
|---|---|---|
| National Institute of General Medical Sciences | R01GM138415 | Chad G Pearson |
| National Institute of General Medical Sciences | R35GM140813 | Chad G Pearson |
| National Science Foundation | 1553798 | Cayla E Jewett |
| National Institutes of Health | INCLUDE T32 GM008730 | Cayla E Jewett |
| Linda Crnic Institute for Down Syndrome, University of Colorado School of Medicine, Anschutz Medical Campus | Blumenthal Fellowship | Cayla E Jewett Bailey L McCurdy |
| National Institutes of Health | R01DK064380 | Rytis Prekeris |
| Global Down Syndrome Foundation, Anna and John Sie Foundation | | Cayla E Jewett Bailey L McCurdy Joaquín M Espinosa Kelly D Sullivan Chad G Pearson |

The funders had no role in study design, data collection and interpretation, or the decision to submit the work for publication.

### Author contributions

Cayla E Jewett, Conceptualization, Data curation, Formal analysis, Validation, Investigation, Visualization, Methodology, Writing – original draft, Writing – review and editing; Bailey L McCurdy, Formal analysis, Investigation, Visualization, Methodology, Writing – review and editing; Eileen T O'Toole, Data curation, Formal analysis, Visualization, Methodology; Alexander J Stemm-Wolf, Data curation, Formal analysis, Investigation, Methodology; Katherine S Given, Resources, Validation, Methodology; Carrie H Lin, Investigation, Visualization; Valerie Olsen, Formal analysis, Investigation, Visualization; Whitney Martin, Joaquín M Espinosa, Kelly D Sullivan, Resources; Laura Reinholdt, Wendy B Macklin, Resources, Supervision, Methodology, Writing – review and editing; Rytis Prekeris, Supervision, Project administration, Writing – review and editing; Chad G Pearson, Conceptualization, Resources,

Supervision, Funding acquisition, Validation, Writing – original draft, Project administration, Writing – review and editing

## Author ORCIDs
Cayla E Jewett http://orcid.org/0000-0002-8406-0814
Laura Reinholdt http://orcid.org/0000-0003-4054-4048
Joaquín M Espinosa http://orcid.org/0000-0001-9048-1941
Kelly D Sullivan http://orcid.org/0000-0003-2725-0205
Wendy B Macklin http://orcid.org/0000-0002-1252-0607
Rytis Prekeris http://orcid.org/0000-0003-3393-1963
Chad G Pearson http://orcid.org/0000-0003-1915-6593

## Ethics

All procedures involving mice were approved by the Institutional Animal Care and Use Committee (IACUC) at the University of Colorado Anschutz Medical Campus (protocol numbers 134 and 00111) and were performed in accordance with National Institute Guidelines for the care and use of animals in research. Both male and female mice were used for experiments, and we observed no differences between sexes.

## Decision letter and Author response

Decision letter https://doi.org/10.7554/eLife.78202.sa1
Author response https://doi.org/10.7554/eLife.78202.sa2

## Additional files

### Supplementary files
• Transparent reporting form

### Data availability
All data generated or analysed during this study will be included in the manuscript and supporting files.

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

# Appendix 1

## Appendix 1—key resources table

| Reagent type (species) or resource | Designation | Source or reference | Identifiers | Additional information |
|---|---|---|---|---|
| Genetic reagent (*M. musculus*) | Dp16 | Jackson Laboratory | stock# 013530 B6.129S7-Dp(16Lipi-Zbtb21)1Yey/J | PMID:17412756 |
| Genetic reagent (*M. musculus*) | Dp10 | Jackson Laboratory | stock# 013529 B6;129-Dp(10Prmt2-Pdxk)2Yey/J | PMID:20442137 |
| Genetic reagent (*M. musculus*) | Dp17 | Jackson Laboratory | stock# 013531 B6;129-Dp(17Abcg1-Rrp1b)3Yey/J | PMID:20442137 |
| Cell line (*Homo-sapiens*) | RPE1 D21 | Gift from Andrew Lane | | PMID:24747640 |
| Cell line (*Homo-sapiens*) | RPE1 T21 | Gift from Andrew Lane | | PMID:24747640 |
| Cell line (*Homo-sapiens*) | RPE1 Q21 | Gift from Andrew Lane | | PMID:24747640 |
| Antibody | PCNT (Rabbit polyclonal) | Abcam | Cat# AB4448, RRID:AB_304461 | IF (1:2000) |
| Antibody | DM1A (mouse monoclonal) | Sigma | Cat# T6199, RRID:AB_477583 | IF (1:300) |
| Antibody | GT335 (mouse monoclonal) | Adipogen | Cat# AG-20B-0020-C100, RRID:AB_2490210 | IF (1:500) |
| Antibody | MYO5A (rabbit polyclonal) | Novus Biologicals | Cat# NBP1-92156, RRID:AB_11017070 | IF (1:500) WB (1:1000) |
| Antibody | CEP164 (rabbit polyclonal) | Protein Tech | Cat# 22227–1-AP, RRID:AB_2651175 | IF (1:500) |
| Antibody | ODF2 (rabbit polyclonal) | Sigma | Cat# HPA001874 RRID:AB_1079522 | IF (1:200) |
| Antibody | Centrin (mouse monoclonal) | Sigma | Cat# 04–1624, RRID:AB_10563501 | IF (1:500) |
| Antibody | Ninein (rabbit polyclonal) | Protein Tech | Cat# 67132–1-Ig, RRID:AB_2882431 | IF (1:200) |
| Antibody | Actub (mouse monoclonal) | Sigma | Cat# T7451, RRID:AB_609894 | IF (1:1000) |
| Antibody | CP110 (rabbit polyclonal) | Protein Tech | Cat# 12780–1-AP, RRID:AB_10638480 | IF (1:500) |
| Antibody | γ-tubulin (DQ19) (rabbit polyclonal) | Sigma | Cat# T3195, RRID:AB_261651 | IF (1:500) |
| Antibody | RAB8 (mouse monoclonal) | BD Transduction Laboratories | Cat# 610844, RRID:AB_398163 | IF (1:100) |
| Antibody | CEP97 (rabbit polyclonal) | Protein Tech | Cat# 22050–1-AP, RRID:AB_11182378 | IF (1:1000) |
| Antibody | GM130 (mouse monoclonal) | BD Transduction Laboratories | Cat# 610822, RRID:AB_398141 | IF (1:100) |
| Antibody | Golgin97 (mouse monoclonal) | Invitrogen | Cat# A21270, RRID:AB_221447 | IF (1:100) |
| Antibody | EEA1 (rabbit polyclonal) | Gift from A. Peden | | IF (1:100) |
| Antibody | CD63 (mouse monoclonal) | Gift from A. Peden | | IF (1:100) |
| Antibody | CEP290 (rabbit polyclonal) | Bethyl | Cat# A301-659A, RRID:AB_1210910 | IF (1:500) |
| Antibody | RPGRIP1L (rabbit polyclonal) | Protein Tech | Cat# 55160–1-AP, RRID:AB_10860269 | IF (1:200) |

*Appendix 1 Continued on next page*

*Appendix 1 Continued*

| Reagent type (species) or resource | Designation | Source or reference | Identifiers | Additional information |
|---|---|---|---|---|
| Antibody | NPHP4 (rabbit polyclonal) | Protein Tech | Cat# 13812–1-AP, RRID:AB_10640302 | IF (1:200) |
| Antibody | TMEM67 (rabbit polyclonal) | Protein Tech | Cat# 13975–1-AP, RRID:AB_10638441 | IF (1:200) |
| Antibody | GFP (mouse monoclonal) | Life Technologies | Cat# A11120, RRID:AB_221568 | IF (1:1000) |
| Antibody | ARL13B (mouse monoclonal) | NeuroMab | Cat# N295B/66, RRID:AB_2877361 | IF (1:500) |
| Antibody | Actub (rabbit polyclonal) | Cell Signaling | Cat# 5335, RRID:AB_10544694 | IF (1:1000) |
| Antibody | PCNT (mouse monoclonal) | BD Transduction Laboratories | Cat# 611814, RRID:AB_399294 | IF (1:200) |
| Antibody | SMO (rabbit polyclonal) | Gift from R. Rohatgi | | IF (1:500) |
| Antibody | Ki67 (rabbit polyclonal) | Abcam | Cat# AB15580, RRID:AB_443209 | IF (1:500) |
| Antibody | CALB1 (chicken) | Neuromics | Cat# CH22118, RRID:AB_2737107 | IF (1:1000) |
| Antibody | DCX (rabbit polyclonal) | Abcam | Cat# AB18723, RRID:AB_732011 | IF (1:500) |
| Antibody | Pan-neuronal marker (rabbit polyclonal) | Sigma | Cat# ABN2300C3, RRID:AB_10953180 | IF (1:100) |
| Antibody | Alexa 488 Anti-Rabbit secondary | Jackson ImmunoResearch | Cat#711-545-152 | IF (1:500) |
| Antibody | Alexa 594 Anti-Rabbit secondary | Jackson ImmunoResearch | Cat#711-585-152 | IF (1:500) |
| Antibody | Alexa 488 Anti-Mouse secondary | Jackson ImmunoResearch | Cat#711-545-150 | IF (1:500) |
| Antibody | Alexa 594 Anti-Mouse secondary | Jackson ImmunoResearch | Cat#711-585-150 | IF (1:500) |
| Antibody | Alexa 488 Anti-Mouse IgG2a secondary | Invitrogen | Cat#A-21131 | IF (1:500) |
| Antibody | Alexa 488 Anti-Mouse IgG1 secondary | Invitrogen | Cat#A-21121 | IF (1:500) |
| Antibody | Alexa 568 Anti-Mouse IgG1 secondary | Invitrogen | Cat#A-21124 | IF (1:500) |
| Antibody | Alexa 568 Anti-Mouse IgG2b secondary | Invitrogen | Cat#A-21144 | IF (1:500) |
| Antibody | Alexa 647 Anti-Rabbit secondary | Invitrogen | Cat#A-21245 | IF (1:500) |
| Chemical compound | Hoechst 33342 | ThermoFisher Scientific | Cat#62249 | IF (1:2000) |
| Transfected construct | GFP-EHD1 (lentiviral plasmid) | Gift from C. Westlake | | PMID:25686250 |
| Transfected construct | pH-Smoothened (lentiviral plasmid) | Gift from D. Toomre | | PMID:27493724 |
| Sequence-based reagent | Dp10For: | Jackson Laboratory | Genotyping PCR primers | GGCGAACGTGGCGAGAAA |
| Sequence-based reagent | Dp10Rev | Jackson Laboratory | Genotyping PCR primers | CCTGCTGCCAAGCCATCAG |
| Sequence-based reagent | Dp16For | Jackson Laboratory | Genotyping PCR primers | CTGCCAGCCACTCTAGCTCT |
| Sequence-based reagent | Dp16Rev | Jackson Laboratory | Genotyping PCR primers | AATTTCTGTGGGGCAAAATG |
| Sequence-based reagent | Dp17For | Jackson Laboratory | Genotyping PCR primers | GGAGCCAGGGCTGATGGT |

*Appendix 1 Continued on next page*

*Appendix 1 Continued*

| Reagent type (species) or resource | Designation | Source or reference | Identifiers | Additional information |
|---|---|---|---|---|
| Sequence-based reagent | Dp17Rev | Jackson Laboratory | Genotyping PCR primers | CAACGCGGCCTTTTTACG |
| Sequence-based reagent | Cux2For | This paper | Genotyping PCR primers | GGGACATCACCCACCGGTAATCTC |
| Sequence-based reagent | Cux2Rev | This paper | Genotyping PCR primers | GACCACTGAGTCTGGCAACACG |
| Sequence-based reagent | Gli1 F | This paper | RT-PCR primers | GAATTCGTGTGCCATTGGGG |
| Sequence-based reagent | Gli1 R | This paper | RT-PCR primers | TGGGATCTGTGTAGCGCTTG |
| Sequence-based reagent | Pcna F | This paper | RT-PCR primers | GCACGTATATGCCGAGACCT |
| Sequence-based reagent | Pcna R | This paper | RT-PCR primers | GTAGGAGACAGTGGAGTGGC |
| Sequence-based reagent | siControl | Sigma | Cat #: SIC001-1NMOL | siRNA: human universal negative control #1 |
| Sequence-based reagent | siPCNT | Dharmacon | Cat #: M-012172-01-0005 | siRNA: human PCNT siRNA Smart Pool |
| Sequence-based reagent | siControl | Sigma | Cat #: SIC001-1NMOL | siRNA: mouse Accell Non-targeting Pool |
| Sequence-based reagent | siPCNT | Dharmacon | Cat #: 18541 | siRNA: mouse Accell Pcnt SMARTpool |
| Commercial assay or kit | Lipofectamine RNAi MAX | ThermoFisher Scientific | Cat. #: 13778100 | For RNAi |
| Commercial assay or kit | Lipofectamine 2000 | Invitrogen | Cat. #: 11668–027 | For lentivirus transductions |
| Commercial assay or kit | Antibody labeling kit | Invitrogen | Cat. #: A20181 | For directly conjugating PCNT to Alexa 488 |

