## [Editor Report]

The authors use human trisomy and tetrasomy 21 cell lines and a mouse model to examine the effects of an additional copy of Pericentrin (PCNT) on cell biology, with a focus on ciliation and ciliary Hedgehog signaling. They demonstrate that modestly increased PCNT levels can attenuate ciliogenesis and may result in trisomy 21-associated phenotypes such as cerebellar growth defects. This work advances our understanding of the trafficking defects caused by increased PCNT and has important implications for our understanding of the cellular basis of trisomy 21, a major hereditary human disorder.

---

## [Decision Letter]

**Decision letter after peer review:**

Thank you for submitting your article "Trisomy 21 induces pericentrosomal crowding disrupting early stages of primary ciliogenesis and mouse cerebellar development" for consideration by *eLife*. Your article has been reviewed by 3 peer reviewers, including Jeremy F Reiter as Reviewing Editor and Reviewer #1, and the evaluation has been overseen by Anna Akhmanova as the Senior Editor.

The reviewers have discussed their reviews with one another, and the Reviewing Editor has drafted this to help you prepare a revised submission. We apologize that it took us some time to complete our review.

Essential revisions:

1) Please attend to the apparent discrepancy between Figure S3B, C indicating that centrosomal MYO5A levels are decreased with increasing chromosome 21 ploidy, and Figure 3E indicating that centrosomal MYO5A levels are increased with increasing chromosome 21 ploidy.

2) Please assess whether HH-dependent transcriptional responses, such as Gli1 induction, were reduced in the cerebella of Dp10 mice. Also, please assess whether there are defects in cerebellar morphology at P21 or in the adult Dp10 mice.

3) Please discriminate between delayed and attenuated ciliogenesis in T21 and Q21 cells by assessing whether the frequencies of cilia in the T21 or Q21 cells eventually reach the same level as D21 cells.

*Reviewer #1 (Recommendations for the authors):*

Is the PCNT copy number in the diploid, trisomy, and tetrasomy 21 retinal pigment epithelium cells validated somewhere? Do the authors do a PCNT protein level scale with copy number in these cells? It is difficult to use immunofluorescence to evaluate protein levels. Can the authors simply use Western blot to evaluate relative protein levels? Similar issues pertain to MYO5A levels.

Figures S3B, and C indicate that centrosomal MYO5A levels were decreased with increasing chromosome 21 ploidy. Figure 3E indicates that centrosomal MYO5A levels were increased with increasing chromosome 21 ploidy. How do these results square? Are there different definitions of centrosomal being used? The figure legend for figure 3E is unclear on the time point that is being analyzed.

The authors indicate that SMO levels in response to SAG are reduced with chromosome 21 trisomy. Are transcriptional responses, such as Gli1 induction, similarly reduced? Although there is a morphological defect in the cerebella of Dp10 mice, there is no decrease in EGL proliferation at P4, inconsistent with a marked effect on Hedgehog signaling. Similarly, is Gli1 expression reduced in Dp10 mice cerebella, as would be predicted?

*Reviewer #2 (Recommendations for the authors):*

In Figure 1B and the data in 1D, was there some normalization applied to microtubule staining? For this experiment to make sense, the intensity of labeling of a single microtubule should be the same across all samples (or be subjected to normalization). In the images shown, the intensity values of what appear to be individual microtubules are very different. Also, the models derived from EM in Figure 2D don't seem to show the same microtubule effect, as noted in the text (line 173). The explanation for this – that the microtubules must be further out from the centrosome – seems weak.

Line 140, Figure 1E, F. It seems from the figure legend that the presence of a cilium was assessed by DM1a staining. It is hard to see how that could be a definitive assay for a cilium given the staining shown in Figure 1B. Are there data using Arl13B or some other cilium protein?

Figure 2A and Line 552 "Percentages represent cells with indicated marker for 3 N's" It seems that the N values are in a separate supplementary table, which makes it harder to interpret the data than necessary. Could the legend simply state that xx% of cells (n=yy) had the particular labeling in question?

Line 229: "increase in EHD1 that is caught up in the pericentrosomal region" – the words "caught up" should be removed here since all that is really known is that there is more there.

Figure 4 C, D: How many cells were imaged by EM tomography? If it was single cells, what is the evidence that these are representative?

Figure 4E. I'm not sure that the cartoon adds much to the clarity of the work. It is harder to discern what details matter in the cartoon than it is to state them in words.

Figure 6G. The SMO localization defect in Dp10 is striking. This was after 4 h of SAG treatment. Did longer treatment result in SMO accumulation?

---

## [Author Response]

Essential revisions:1) Please attend to the apparent discrepancy between Figure S3B, C indicating that centrosomal MYO5A levels are decreased with increasing chromosome 21 ploidy, and Figure 3E indicating that centrosomal MYO5A levels are increased with increasing chromosome 21 ploidy.

Figure 3E, shows MYO5A levels at 24h in a 0.0-1.2 µm region from the centroid of the centrosome. Figure S3B and S3C (now S3C and S3D) show a similar analysis but for 2h and 4h post serum depletion. Therefore, Figure 3E can only be compared to Figure S3E (24h post serum depletion), which also demonstrates similar slightly elevated centrosomal MYO5A levels. We have modified the text and the legends to make this point clearer.

2) Please assess whether HH-dependent transcriptional responses, such as Gli1 induction, were reduced in the cerebella of Dp10 mice. Also, please assess whether there are defects in cerebellar morphology at P21 or in the adult Dp10 mice.

We now show that Gli1 induction is reduced in SAG treated MEFs from Dp10 mice compared to WT mice. We were unable to antibody stain for Gli1 using antigen retrieval and immunofluorescence in the cerebella of P4 WT and Dp10 brains. Unlike P4 mice, the cerebellar ciliation and molecular and granule layer widths were not different in P21 mice. This suggests that ciliogenesis delays are found in the early and rapidly dividing granule cells of P4 but not in later stages of development when cells are not dividing as rapidly. These results are now included in Figures S6 and S7.

3) Please discriminate between delayed and attenuated ciliogenesis in T21 and Q21 cells by assessing whether the frequencies of cilia in the T21 or Q21 cells eventually reach the same level as D21 cells.

We have now performed a ciliation time course in RPE1 D21, T21, and Q21 cells over 7 days. Our new data confirms that increasing HSA21 dosage delays but does not abolish ciliogenesis (Figure S1H).

Reviewer #1 (Recommendations for the authors):Is the PCNT copy number in the diploid, trisomy, and tetrasomy 21 retinal pigment epithelium cells validated somewhere? Do the authors do a PCNT protein level scale with copy number in these cells? It is difficult to use immunofluorescence to evaluate protein levels. Can the authors simply use Western blot to evaluate relative protein levels? Similar issues pertain to MYO5A levels.

We previously validated these cell lines via Western blot and RNA FISH (see PMID: 35476505). The text has been modified to include this. We have now included Western blot analysis of MYO5A levels demonstrating that, like our immunofluorescence results, they are unchanged across D21, T21, and Q21 lines (Figure S3B).

Figures S3B, and C indicate that centrosomal MYO5A levels were decreased with increasing chromosome 21 ploidy. Figure 3E indicates that centrosomal MYO5A levels were increased with increasing chromosome 21 ploidy. How do these results square? Are there different definitions of centrosomal being used? The figure legend for figure 3E is unclear on the time point that is being analyzed.

Figure 3E, shows MYO5A levels at 24h in a 0.0-1.2 µm region from the centroid of the centrosome. Figure S3B and S3C (now S3C and S3D) show the same analysis but for 2h and 4h post serum depletion. Therefore, Figure 3E can only be compared to Figure S3E (24h post serum depletion), which also demonstrates similar slightly elevated MYO5A levels. We have modified the text and the legends to make this point clear.

The authors indicate that SMO levels in response to SAG are reduced with chromosome 21 trisomy. Are transcriptional responses, such as Gli1 induction, similarly reduced? Although there is a morphological defect in the cerebella of Dp10 mice, there is no decrease in EGL proliferation at P4, inconsistent with a marked effect on Hedgehog signaling. Similarly, is Gli1 expression reduced in Dp10 mice cerebella, as would be predicted?

We now show that Gli1 induction is reduced in SAG treated MEFs from Dp10 mice compared to WT mice. We were unable to antibody stain for Gli1 using antigen retrieval and immunofluorescence in the cerebella of P4 WT and Dp10 brains. Unlike P4 mice, the cerebellar ciliation and molecular and granule layer widths were not different in P21 mice. This suggests that ciliogenesis delays are found in the early and rapidly dividing granule cells of P4 but not in later stages of development when cells are not dividing as rapidly. These results are now included in Figures S6 and S7.

Reviewer #2 (Recommendations for the authors):In Figure 1B and the data in 1D, was there some normalization applied to microtubule staining? For this experiment to make sense, the intensity of labeling of a single microtubule should be the same across all samples (or be subjected to normalization). In the images shown, the intensity values of what appear to be individual microtubules are very different. Also, the models derived from EM in Figure 2D don't seem to show the same microtubule effect, as noted in the text (line 173). The explanation for this – that the microtubules must be further out from the centrosome – seems weak.

Figure 1B shows images that were acquired with identical microscope settings and levelled the same. No normalization has been applied. We did this to illustrate the substantial microtubule networks that are recruited to and rearranged around the centrosome prior to ciliogenesis in D21, T21, and Q21 cells.

There are several reasons why we may not see as many microtubules in T21 and Q21 cells by EM compared to IF. First, EM is limited by small sample size and small analysis region. Performing microtubule tracing is technical and time consuming. In addition, longer microtubules are easier to pick out and trace compared to shorter microtubules that would be increasingly present in T21 and Q21 cells (PMID: 35476505). Second, the fixation technique is different for EM (high pressure freeze substitution) compared to IF (PFA with pre-permeabilization step). We addressed the issue of microtubule preservation and stability throughout different fixation techniques in our previous paper (PMID: 35476505).

Line 140, Figure 1E, F. It seems from the figure legend that the presence of a cilium was assessed by DM1a staining. It is hard to see how that could be a definitive assay for a cilium given the staining shown in Figure 1B. Are there data using Arl13B or some other cilium protein?

We have performed the same analyses with ARL13B, Acetylated tubulin, and GT335 as alternative markers for cilia and get similar numbers as with DM1a staining. Using these established markers, we are confident in our ability to identify cilia and differentiate these structures from cytoplasmic microtubules with DM1a staining.

Figure 2A and Line 552 "Percentages represent cells with indicated marker for 3 N's" It seems that the N values are in a separate supplementary table, which makes it harder to interpret the data than necessary. Could the legend simply state that xx% of cells (n=yy) had the particular labeling in question?

We would prefer to keep statistical information in the supplementary table so that all information can be found together and clearly described. However, if required we are willing to add the information to figure legends.

Line 229: "increase in EHD1 that is caught up in the pericentrosomal region" – the words "caught up" should be removed here since all that is really known is that there is more there.

We agree and have removed this phrase from the text.

Figure 4 C, D: How many cells were imaged by EM tomography? If it was single cells, what is the evidence that these are representative?

A total of 20 D21 cells were screened, and 4 complete centrosomes were selected for tomographic reconstruction. A total of 26 T21 cells were screened, and 3 complete centrosomes were selected for tomographic reconstruction. A total of 17 Q21 cells were screened and 4 were chosen for reconstruction. Only those cells that had the complete centrosome in the 3D volume of the serial sections were selected for reconstruction. However, observations about the presence of ciliary vesicles and primary cilia could be made from the incomplete centrosomes. This is described in the Methods and we have now included it in the figure legends.

Figure 4E. I'm not sure that the cartoon adds much to the clarity of the work. It is harder to discern what details matter in the cartoon than it is to state them in words.

We have removed the cartoon from the figure.

Figure 6G. The SMO localization defect in Dp10 is striking. This was after 4 h of SAG treatment. Did longer treatment result in SMO accumulation?

It would be interesting to examine SMO localization after longer SAG treatment; however, we did not try this experiment due to limited resources and published protocols demonstrating that the majority of SMO translocation occurs within the first 4h of SAG treatment (PMID: 17641202). We anticipate that our future studies will test this question.